# MSMR: BANDIT WITH MINIMAL SWITCHING COST AND MINIMAL MARGINAL REGRET

## ABSTRACT

Effectively balancing switching costs and regret remains a fundamental challenge in bandit learning, especially when the arms exhibit similar expected rewards. Traditional upper confidence bound (UCB) -based algorithms struggle with this trade-off by frequently switching during exploration, incurring high cumulative switching costs. Recent approaches attempt to reduce switching by introducing structured exploration or phase-based selection, yet they often do so at the expense of increased regret due to excessive exploitation of suboptimal arms. In this paper, we propose a new unified framework for bandit problems with switching costs, containing several classical algorithms, applicable to both Multi-Armed Bandits (MAB) and Combinatorial Multi-Armed Bandits (CMAB). Our approach is built on three key components: initial concentrated exploration, near-optimal exploitation, and predictive selection, which together achieve a principled balance between switching cost and regret. Based on this framework, we introduce the Minimal Switching Cost and Minimal Marginal Regret (MSMR) family of algorithms. Theoretically, we show that MSMR algorithms achieve a regret upper bound of $\mathcal{O}(\log n)$ over horizon $n$, incur only $\mathcal{O}((\log n)^{1-\varepsilon})$ switching cost, and its marginal loss has an upper bound of $\mathcal{O}(\lambda\sqrt{\log n})$ by setting $\varepsilon = 1/2$, where $\lambda$ and $\varepsilon \in (0, 1)$ are hyper-parameters. Experiments show that MSMR algorithms reduce switching costs to 1.0% (MAB) and 1.3% (CMAB) of those incurred by standard baselines, while maintaining comparable regret, demonstrating their practical effectiveness.

## 1 INTRODUCTION

The stochastic multi-armed bandit (MAB) problem is a classical framework for sequential decision-making, where a learning agent repeatedly chooses from a set of arms with unknown reward distributions to minimize cumulative regret (16). Over the years, this framework has been extended to accommodate more complex scenarios. A notable generalization is the combinatorial multi-armed bandit (CMAB) problem, where the agent selects a subset of base arms—known as a super arm—in each round, and receives feedback from all the selected base arms. This formulation captures a wide range of real-world applications, including online advertising, network optimization, and healthcare systems (6; 25; 21; 19; 9).

To minimize regret, traditional bandit algorithms aim to balance exploration and exploitation. Since Thompson's early work on bandits for clinical trials (24), a rich body of theoretical and algorithmic developments has followed. Lai and Robbins (15) established the first lower bounds for regret, showing that it must grow at least logarithmically in the number of rounds. Auer et al. (3) proposed the UCB algorithm, achieving logarithmic regret. In the combinatorial setting, Chen et al. (6) introduced the CUCB algorithm, which was later extended to probabilistically triggered arms (8), both achieving $\mathcal{O}(\log n)$ regret bounds.

However, in many practical applications, regret is not the only performance measure, as switching between arms across rounds may incur explicit or implicit costs. For example, in co-branding recommendation systems,

Table 1: Comparison of Regret, Switching Cost, and Marginal Loss Upper Bound.

| Setting | MAB Algorithm | Regret | Switching Cost | Marginal Loss |
|---|---|---|---|---|
| MAB | UCB(3) | $\mathcal{O}(\log n)$ | $\mathcal{O}(\log n)$ | $\mathcal{O}(\lambda \log(n))$ |
|  | Phased-UCB(16) | $\mathcal{O}(\log n)$ | $\mathcal{O}(\log \log n)^{*}$ | $\mathcal{O}(\log n + \lambda \log \log n)$ |
|  | Batched Tsallis-INF(1) | $\mathcal{O}(\log n)$ | —— | $\mathcal{O}(\lambda^{1/3} n^{2/3})$ |
|  | Batched Arm Elimination (13) | $\mathcal{O}(n^{1/B} \log n)^{**}$ | $\mathcal{O}(B)$ | $\mathcal{O}(n^{1/B} \log n + \lambda B)$ |
|  | MSMR-UCB | $\mathcal{O}((\log n)^{\varepsilon})$ | $\mathcal{O}((\log n)^{1-\varepsilon})$ | $\mathcal{O}(\lambda \sqrt{\log n})$ |
| CMAB | CUCB(6) | $\mathcal{O}(\log n)$ | $\mathcal{O}(\log n)$ | $\mathcal{O}(\lambda \log n)$ |
|  | Phased-CUCB(16) | $\mathcal{O}(\log n)$ | $\mathcal{O}(\log \log n)$ | $\mathcal{O}(\log n + \lambda \log \log n)$ |
|  | B-FTRL (11) | $\mathcal{O}(n^{2/3})$ | $\mathcal{O}(n^{2/3})$ | $\mathcal{O}(\lambda n^{2/3})$ |
|  | MSMR-CUCB | $\mathcal{O}((\log n)^{\varepsilon})$ | $\mathcal{O}((\log n)^{1-\varepsilon})$ | $\mathcal{O}(\lambda \sqrt{\log n})$ |

$^{*}$ Appendix J theoretically analyzes why MSMR performs better than Phased methods.
$^{**}$ $B$ is the number of batches and small $B$ will cause large regret.

repeatedly switching recommended items adds fixed overhead beyond suboptimal choices. Similarly, in session-based recommendation scenarios (26), frequent product changes can fragment user attention and reduce click-through rates (CTR), creating additional operational costs. These overheads are commonly termed *switching costs*. To address this, several recent works propose switching-aware bandit algorithms. A common approach uses phased strategies that repeatedly select the same (super) arm within each phase, limiting the number of switches. This framework has been applied in both MAB settings, such as Batched Tsallis-INF (16; 14; 22; 1), and CMAB settings, such as B-FTRL (11), to reduce switching frequency.

Despite these advances, existing methods still suffer from two fundamental dilemmas: **(1) The dilemma between regret and switching cost:** To reduce switching, current algorithms often tolerate increased regret, both Batched Tsallis-INF and B-FTRL have polynomial-level regret, which may be unacceptable in regret-sensitive applications. **(2) The dilemma among arms with similar rewards:** When many (super) arms have similar expected rewards, existing methods like B-FTRL tend to oscillate between them, resulting in excessive switching while elimination-based methods (14) risk converging to suboptimal policy.

Regarding these challenges, we propose a novel framework called Bandit with Minimal Switching Cost and Minimal Marginal Regret (MSMR), which incorporates three key technical modules: *initial concentrated exploration*, *near-optimal exploitation*, and *predictive selection*. The initial concentrated exploration phase occurs at the beginning of the learning process and uses a single phase to gathering sufficient information for each arm. The near-optimal exploitation technique determines whether the currently selected arm should be pulled additional times within the current phase. The predictive selection technique anticipates whether the currently selected arm will need to be explored in the near future, allowing the algorithm to explore it proactively in advance. We prove the effectiveness of these techniques and theoretically demonstrate that MSMR achieves a switching cost of only $\mathcal{O}((\log n)^{\varepsilon})$, while maintaining asymptotically the same regret as standard bandit algorithms, which is $\mathcal{O}(\log n)$, where $n$ is the time horizon and $0 < \varepsilon < 1$ is a hyper-parameter we can choose flexibly. The main contributions of this paper are as follows:

- We propose a novel unified framework to address the two dilemmas in bandit problems: the trade-off between regret and switching cost, and the instability caused by arms with similar expected rewards. This framework, which encompasses a range of classical algorithms, incorporates three core techniques and is highly flexible, allowing it to adapt to a wide range of bandit settings, including both MAB and CMAB.

- We provide rigorous theoretical guarantees for each core technique in the framework and prove that MSMR algorithms achieve a significantly improved trade-off between regret and switching cost. Unlike existing methods that typically reduce switching cost at the expense of marginal regret, through the carefully designed exploitation function in the *near-optimal exploitation module*, MSMR asymptotically achieves the same regret as standard algorithms, while incurring only minimal switching cost.

- We conduct extensive experiments and ablation studies on MSMR algorithms. The results show that MSMR achieves only 1.0% and 1.3% of the switching cost incurred by standard methods in MAB and CMAB settings respectively, while maintaining nearly the same level of regret. These results highlight the superior performance of our framework and the effectiveness of the key techniques.

## 2 RELATED WORKS

The Multi-Armed Bandit (MAB) problem serves as a foundational model in sequential decision-making, balancing exploration and exploitation to minimize regret (16). While MAB focuses on selecting a single arm, real-world applications like online advertising often require choosing combinations of arms, leading to the Combinatorial Multi-Armed Bandit (CMAB) framework, which generalizes MAB by allowing the selection of super arms, i.e., combinations of base arms, at each round (8; 25; 20).

**Phased Bandits:** Phased bandit algorithms partition the learning process into discrete phases, maintaining a fixed action within each phase to reduce computational overhead and accelerate exploration. This approach mitigates the frequent updates required in traditional bandit algorithms, offering efficiency gains. (16) provide a comprehensive overview of bandit algorithms, including phased strategies. In the batched setting, Perchet et al. (2016) analyze the trade-offs between batch size and regret, demonstrating that appropriately chosen batch sizes can yield near-optimal performance. In adversarial contexts, (11) introduce algorithms that adaptively determine phase lengths to balance exploration and exploitation effectively. Moreover, (4) discusses the benefits of structured exploration in adversarial environments. These phased approaches are particularly beneficial in scenarios where switching costs or computational constraints are significant concerns.

**Switching Cost:** Incorporating switching costs into bandit problems introduces additional complexity, as learners must balance the trade-off between exploration benefits and the incurred costs of changing actions. (11) analyze this scenario, establishing a regret lower bound of $\widetilde{\Theta}(n)$ for adversarial bandits with unit switching costs. (23) further explored the stochastic setting, revealing phase transitions in optimal regret rates as a function of the switching budget. Then (14; 1) gave the Batched Tsallis methods in MAB setting and (11) B-FTRL in CMAB settings. Phased strategies naturally align with the goal of minimizing switching costs by limiting action changes to phase boundaries. (2) extended this concept to settings with feedback graphs, proposing algorithms that consider both the structure of feedback and switching costs. These approaches demonstrate that structured exploration can effectively manage switching costs without significantly compromising regret.

While existing algorithms address either regret minimization or switching cost reduction, achieving an optimal balance between the two remains challenging. (11) highlights that minimizing switching costs often leads to increased regret, as infrequent action changes can hinder exploration. (23) demonstrates that strict switching constraints can cause abrupt changes in optimal strategies, complicating the learning process. Moreover, in environments with numerous similar arms, algorithms may oscillate between near-optimal actions, incurring unnecessary switching costs without substantial gains in reward. (2) addresses this by incorporating feedback graphs, yet challenges persist in balancing exploration and exploitation under switching constraints. The newest methods given by (14; 1; 11) still cause a large number of regret though reducing switching costs. These limitations highlight the need for a novel framework that significantly reduces switching costs while incurring only minimal marginal regret compared to standard methods.

## 3 PROBLEM SETUP

**Regret.** We denote $[\![K]\!]$ as the set $\{1, 2, \ldots, K\}$ for any $K \in \mathbb{N}^+$, and $\zeta(\cdot)$ as the Riemann Zeta Function, which is $\zeta(s) = \sum_{n=1}^{\infty} n^{-s}$. Let $[\![K]\!]$ denotes the set of arms. For each arm $i \in [\![K]\!]$, pulling it at round $t$ yields a reward feedback $X_{i,t} \in [0, 1]$. The unknown reward vector is represented by $\boldsymbol{\mu} = (\mu_1, \ldots, \mu_K)$,

where $\mu_i = \mathbb{E}[X_{i,t}]$ denotes the expected reward for any arm $i$. The optimal arm is denoted as the arm $i_*$ which maximizes the expected reward, i.e., $\mu_* = \max_{j \in [\![K]\!]} \mu_j$. At each round $t$, the agent selects an arm $i_t$. The objective of the MAB problem is to identify this optimal arm while minimizing regret in time horizon $n$, which is defined as:

$$Reg(n) = n\mu_* - \mathbb{E}[\sum_{t=1}^{n} X_{i_t,t}]. \tag{1}$$

Based on the definition above, in the CMAB scenario, the learning agent selects a combination of multiple base arms from $[\![K]\!]$ at each round, referred to this combination as a super arm $S$, which has $m$ base arm in it. Let $\mathcal{S}$ as the set of all feasible super arms. At each round $t$, the agent selects a super arm $S_t \in \mathcal{S}$, and the outcomes $X_{i,t}$ for all base arms $i$ in $S_t$ are revealed. The reward for a selected super arm $S_t$ at round $t$, denoted as $R(S_t)$, is a non-negative random variable that depends on the specific problem instance, the selected super arm $S_t$, and the rewards of the revealed base arms. In some scenarios, the reward can be simply as the sum of the rewards of the base arms in $S_t$: $R(S_t) = \sum_{i \in S} X_{i,t}$ (18), while in more general cases, the reward function can be more complex, such as nonlinear functions, non-symmetric functions of rewards from these base arms, etc. The expected reward of selecting a super arm is defined as $r_{\boldsymbol{\mu}}(S) = \mathbb{E}[R(S_t)]$. The optimal super arm is denoted as the super arm $S_*$ that maximizes the expected reward, i.e., $r_{\boldsymbol{\mu}}(S_*) = \max_{S \in \mathcal{S}} r_{\boldsymbol{\mu}}(S)$. The goal of CMAB is to identify the optimal super arm while minimizing regret. For many reward functions, computing the exact $S_*$ is NP-hard, even when $\boldsymbol{\mu}$ is known. To address this, CMAB literature (25; 27; 20; 9) often assumes access to an offline $(\alpha, \beta)$-approximation oracle. This oracle, for given parameters $\alpha, \beta \leq 1$, takes an expectation vector $\boldsymbol{\mu}$ as input, and outputs a super arm $S \in \mathcal{S}$, such that $P[r_{\boldsymbol{\mu}}(S) \geq \alpha \cdot \text{opt}_{\boldsymbol{\mu}}] \geq \beta$, where $\beta$ is the success probability of the oracle, and $\text{opt}_{\boldsymbol{\mu}} = r_{\boldsymbol{\mu}}(S_*)$ is the mean reward of the optimal super arm. The $(\alpha, \beta)$-approximation regret of a CMAB algorithm after $n$ rounds of play using such an oracle under the expectation vector $\boldsymbol{\mu}$ is formally defined as:

$$Reg_{\boldsymbol{\mu}, \alpha, \beta}(n) = n \cdot \alpha \cdot \beta \cdot \text{opt}_{\boldsymbol{\mu}} - \mathbb{E}\left[\sum_{t=1}^{n} r_{\boldsymbol{\mu}}(S_t)\right]. \tag{2}$$

Following (6; 8; 20), we make two mild assumptions about the expected reward $r_{\boldsymbol{\mu}}(S)$:

- **Monotonicity**. The expected reward of playing any super arm $S \in \mathcal{S}$ is monotonically non-decreasing with respect to the expectation vector. Specifically, if for all $i \in [\![K]\!]$, $\mu_i \leq \mu_i'$, then $r_{\boldsymbol{\mu}}(S) \leq r_{\boldsymbol{\mu}'}(S)$ for all $S \in \mathcal{S}$.

- **Bounded smoothness**. There exists a strictly increasing (and thus invertible) function $f(\cdot)$, called the bounded smoothness function, such that : (1) for any two super arm $S$ and $S'$, we have $0 \leq r_{\boldsymbol{\mu}}(S) - r_{\boldsymbol{\mu}}(S') \leq f(\Gamma_1)$ if $\min_{i \in S} \max_{j \in S'} |\mu_i - \mu_j| \leq \Gamma_1$ . (2) for any two expectation vectors $\boldsymbol{\mu}$ and $\boldsymbol{\mu}'$, we have $|r_{\boldsymbol{\mu}}(S) - r_{\boldsymbol{\mu}'}(S)| \leq f(\Gamma_2)$ if $\max_{i \in S} |\mu_i - \mu_i'| \leq \Gamma_2$.

**Switching Cost.** We define the *switching cost* as the total number of times the agent changes its selected (super) arm between consecutive rounds. Formally, the switching cost $C(n)$ in MAB and CMAB settings are given by:

$$C(n) = \mathbb{E}[\sum_{t=1}^{n-1} \mathbb{I}(i_t \neq i_{t+1})], \quad \text{and} \quad C(n) = \mathbb{E}[\sum_{t=1}^{n-1} \mathbb{I}(S_t \neq S_{t+1})], \tag{3}$$

where $\mathbb{I}(\cdot)$ is the indicating function. This metric quantifies switching costs between selecting different (super) arms and is critical in applications where frequent changes incur penalties (11; 22).

**Marginal Loss.** When evaluating the trade-off between regret and switching cost, some studies(1; 22) have adopted a linear combination of the two as an integrated performance metric. Following a similar approach, this paper defines the marginal loss as the difference between such a combination and its theoretically optimal counterpart, formally given as:

$$R^A(\lambda, n) = (Reg^A(n) - Reg^{opt}(n)) + \lambda(C^A(n) - C^{opt}(n)), \tag{4}$$

where $Reg^{opt}(n)$ is the lowest regret that can be achieved up to now, $C^{opt}(n)$ is the lowest switching cost that can be achieved up to now. We choose standard UCB(CUCB) algorithm as $Reg^{opt}(n)$ and greedy algorithm as $C^{opt}(n)$(see Appendix G for details). In this setting, the metric reflects the capability of balancing regret and switching cost by comparing the existing measure with its theoretical optimum.

**Upper Confidence Bound.** In bandit problems, Upper Confidence Bound (UCB)-based approaches are widely used to balance exploration and exploitation. These methods aim to exploit the best-known (super) arms while still exploring less-visited ones to avoid convergence to suboptimal solutions (16). Specifically, they maintain an upper confidence estimate for each (super) arm that combines its empirical mean with an exploration bonus, and select the (super) arm with the highest estimate at each round. The width of the confidence interval controls the level of exploration (19). Below, we outline the design of UCB in both the MAB and CMAB settings.

In the MAB setting, let $T_i(t)$ denote the number of times arm $i$ has been pulled up to round $t$. The empirical mean reward of arm $i$ is given by: $\hat{\mu}_{i,t} = (1/T_i(t)) \sum_{s=1}^{t} X_{i,s} \cdot \mathbb{I}(j_s = i)$. The corresponding upper confidence estimate is: $\bar{\mu}_{i,t} = \hat{\mu}_{i,t} + c_{i,T_i(t)}$, where the confidence bonus is defined as $c_{i,T_i(t)} = \sqrt{2 \ln t / T_i(t)}$. In the CMAB setting, $T_i(t)$ and $\hat{\mu}_{i,t}$ are computed in the same way as in the MAB case. The upper confidence estimate is also defined as $\bar{\mu}_{i,t} = \hat{\mu}_{i,t} + c_{i,T_i(t)}$, but with a slightly different confidence interval: $c_{i,T_i(t)} = \sqrt{3 \ln t / 2T_i(t)}$, since the super arm involves the combination of multiple base arms. For specific derivations, please refer to (6).

## 4 ALGORITHMS

Determining "*when to switch without sacrificing performance*" is a central challenge in bandit learning with switching costs. To address the trade-off between regret and switching cost, we propose the Bandit with Minimal Switching Cost and Minimal Marginal Regret (MSMR) framework for various bandit settings. MSMR begins with an "Initial Concentrated Exploration" phase, where all arms are explored in a single batch to collect sufficient statistics while avoiding repeated switches. In the subsequent "Near-optimal Exploitation" phase, switching costs are reduced by favoring arms with higher empirical rewards. Together, these phases substantially lower switching costs while incurring only minimal marginal regret. To further improve performance when (super) arms have similar rewards, we introduce a "Predictive Selection" technique that anticipates near-future selections to prevent unnecessary switches. Due to space constraints, we only present MSMR-CUCB in the main text and other variant is provided in Appendix C.

**Phase 1:** We adopt an initial concentrated exploration strategy. In the MAB setting, we continue exploring an arm $i$ as long as $\hat{\mu}_{i,t} + \sqrt{2 \ln n / T_i(t)} \geq 1$. In the CMAB setting, a super arm $S$ is explored while $\min_{j \in S} \hat{\mu}_{j,t} + \sqrt{3 \ln n / 2T_j(t)} \geq 1$, enabling efficient, compact exploration. Due to feature of UCB-based algorithms, where the frequency of exploring a suboptimal (super) arm depends on its reward gap from the optimal, even the worst-performing (super) arm is pulled at least $\mathcal{O}(\log n)$ times (15). Building on this, we concentrate these inevitable explorations into a single initial phase. Allocating sufficient exploration at the start naturally reduces switching costs and gathers information crucial for effective learning in subsequent phases.

**Phase 2:** If the selected (super) arm $i$ or $S$ has a relatively large empirical estimate, the agent pulls it $\gamma_i(t)$ or $\gamma_S(t) = \gamma_{\arg\min_{j \in S} T_j(t)}(t)$ times (we express it generally as $\gamma(t)$ for simplicity; see Section 5 for detailed expressions), where $\gamma(\cdot)$ is the exploitation function. For example, $\gamma(t) = 1$ in CUCB (6), while $\gamma(t) = \mathcal{O}(t^{1/2})$ in B-FTRL. A "relatively large empirical estimate" means $i_t = \hat{i}_{*,t}$ in the MAB setting and $r_{\hat{\mu}}(S_t) \geq \alpha \cdot r_{\hat{\mu}}(\hat{S}_{*,t})$ in the CMAB setting, where $i_t = \arg\max_{j \in [\![K]\!]} \bar{\mu}_{j,t}$, $\hat{i}_{*,t} = \arg\max_{j \in [\![K]\!]} \hat{\mu}_{j,t}$, $S_t = \arg\max_{S \in \mathcal{S}} r_{\bar{\mu}}(S)$, and $\hat{S}_{*,t} = \arg\max_{S \in \mathcal{S}} r_{\hat{\mu}}(S)$ in round $t$. Here, $\alpha$ is the approximation parameter in the $(\alpha, \beta)$-approximation oracle.

Near-optimal exploitation is a strategy where the agent repeatedly exploits a selected (super) arm with a relatively large empirical estimate, assuming it is near-optimal. We argue that when the UCB-selected arm also attains the highest empirical mean, it becomes a more reliable candidate for "near-optimal exploitation," justifying more aggressive exploration. If the selected (super) arm is optimal, the agent exploits it without incurring any regret. If it is suboptimal, its empirical estimate is typically close to the optimal arm's. As noted in (15; 16), when expected rewards are similar, the theoretical lower bound on required explorations increases substantially. To reduce switching costs, these repeated explorations are grouped into a single phase. Importantly, exploring such arms in advance does not significantly affect overall regret, since these steps are inevitable under UCB-based algorithms and mainly occur during the under-sampled stage (see Appendix E for details).

**Predictive Selection:** In bandit problems, switching costs arise from two sources: (1) switching between suboptimal (super) arms and the optimal one, and (2) switching among suboptimal (super) arms. Once the agent selects a suboptimal (super) arm, at least one switching cost with the optimal arm is incurred, which is inevitable under UCB-based algorithms. Consider the MAB setting: if the optimal arm has the lowest upper confidence bound, the agent may keep exploring suboptimal arms until the optimal arm attains the highest bound, resulting in many switches among suboptimal arms. However, if we can predict that the selected arm $i_t$ will be explored in the future before the optimal arm reaches the highest bound, we may explore $i_t$ in advance even if its current estimate is not the highest. This strategy reduces switching among suboptimal arms and applies similarly to the CMAB setting, hence the term *predictive selection*.

In MAB setting, for any arm $i$, once the agent chooses an arm $i_{t_1} \neq \hat{i}_{*,t_1}$ to explore at round $t_1$, the algorithm will repeatedly select $i_{t_1}$ in the subsequent phase regardless of the upper confidence estimation if the following inequality holds:

$$\sqrt{2\ln(t_2)/T_{i_{t_1}}(t_2)} + \hat{\mu}_{i_{t_1},t_2} \geq \sqrt{2\ln(t_3)/T_{\hat{i}_{*,t_2}}(t_2)} + \hat{\mu}_{\hat{i}_{*,t_2},t_2}, \tag{5}$$

where $t_2 = t_1 + \gamma_{i_{t_1}}(t_1)$ and $t_3 = t_2 + \sum_{j \neq i_{t_1}, \hat{i}_{*,t_2}} \gamma_j(t_2)$.

In CMAB setting , once the agent chooses a super arm $S_{t_1} \neq \hat{S}_{*,t_1}$ to explore at round $t_1$, the algorithm will repeatedly select $S_{t_1}$ in the subsequent phase regardless of the upper confidence estimation if the following inequality holds:

$$r(\hat{\boldsymbol{\mu}}_{t_2}, \boldsymbol{c}_{t_2}, S_{t_1}) \geq r(\hat{\boldsymbol{\mu}}_{t_2}, \boldsymbol{c}_{t_3}, \hat{S}_{*,t_2}), \tag{6}$$

where $\boldsymbol{c}_t = (c_{i,t})_{i \in [\![K]\!]}$ , $r(\hat{\boldsymbol{\mu}}_t, \boldsymbol{c}_t, \cdot) \triangleq r_{\hat{\boldsymbol{\mu}}_t + \boldsymbol{c}_t}(\cdot) = r_{\bar{\boldsymbol{\mu}}_t}(\cdot)$, $t_2 = t_1 + \gamma_{S_{t_1}}(t_1)$ and $t_3 = t_2 + \sum_{j \in \{[\![K]\!] \setminus (S_{t_1} \bigcup \hat{S}_{*,t_2})\}} \gamma_j(t_2)$. We use MSMR-P to represents the MSMR Algorithm with Predictive Selection 4 (See Appendix C for MSMR Algorithm with Predictive Selection).

## 5 THEORETICAL ANALYSIS

In this section, we present some theoretical analyses of our proposed methods, including the regret bounds and switching cost of the MSMR algorithms. A comparison between the MSMR algorithms and other existing methods is shown in Table 1.

**Lemma 5.1** *Initial concentrated exploration doesn't increase marginal regret with a probability larger than* $1 - Kn^{-4}$ *in MAB and* $1 - Kn^{-3}$ *in CMAB.*

Lemma 5.1 shows that the MSMR algorithm, when equipped with the *initial concentrated exploration* technique, incurs no higher regret than the standard MSMR algorithm without this technique. As demonstrated in Appendix D, the regret incurred during the initial phase is captured by the under-sampled stage of UCB-based algorithms. Therefore, the *initial concentrated exploration* technique effectively reduces switching costs without introducing marginal regret with a large probability.

---

**Algorithm 1** MSMR-CUCB Algorithm

---

**Input:** Time horizon $n$, constant $M$, $\alpha$, function $\gamma(\cdot)$
1: $t \leftarrow 1$, $\hat{\mu}_i \leftarrow 1$ for all $i$
2: **while** Exists $i$ makes $T_i(t) = 0$ **do**
3:      $\{\hat{\mu}_{j_1}, \hat{\mu}_{j_2}, \ldots, \hat{\mu}_{j_K}\} \leftarrow$ Sort base arm by $\hat{\mu}_i$ in a decreasing way
4:      $S \leftarrow \{j_1, j_2, \ldots, j_m\}$
5:      Play $S$ and observe $X_{i,t}$, for any $i \in S$, update $T_i(t) \leftarrow T_i(t) + 1$, $\hat{\mu}_{i,t} \leftarrow \frac{T_i(t-1)\cdot\hat{\mu}_{i,t-1}+X_{i,t}}{T_i(t)}$,
       $\bar{\mu}_{i,t} \leftarrow \hat{\mu}_{i,t} + \sqrt{3\ln t/2T_i(t)}$, $\bar{\mu}'_{i,t} \leftarrow \hat{\mu}_{i,t} + \sqrt{3\ln n/2T_i(t)}$ until $\min_{j \in S} \bar{\mu}'_{j,t} \leq 1$ and update $t$
6: **end while**
7: **while** $t \leq n$ **do**
8:      $S_t \leftarrow \text{argmax}_{S \in \mathcal{S}} r_{\bar{\mu}}(S)$, $Z \leftarrow \gamma_{S_t}(t)$
9:      $\hat{S}_{*,t} \leftarrow \text{argmax}_{S \in \mathcal{S}} r_{\hat{\mu}}(S)$,
10:     $Z \leftarrow M \cdot \gamma_{S_t}(t)$ when $r_{\hat{\mu}}(S_t) \geq \alpha \cdot r_{\hat{\mu}}(\hat{S}_{*,t})$
11:     Play super arm $S_t \min\{Z, n-t\}$times
12:     Update $t$ and $T_i(t)$, $\hat{\mu}_{i,t}$, $\bar{\mu}_{i,t}$ for all base arms
13: **end while**

---

**Theorem 5.2** *By setting $\gamma_i(t) = N(T_{i_t}(t))^\varepsilon$, $0 < \varepsilon < 1$ and $N$ is a constant, with probability lager than $1 - Kn^{-4}$, the regret upper bound of MSMR-UCB is*

$$\sum_{i \neq i_*} \left( \frac{8\ln n}{\Delta_i} + MN(8\ln(n))^\varepsilon \Delta_i^{1-2\varepsilon} + 2MN \cdot \zeta(2-\varepsilon)\Delta_i \right), \tag{7}$$

*where $\Delta_i = \mu_* - \mu_i$ for each arm $i$.*

Before presenting the regret upper bound of MSMR-CUCB, we first define the gap between super arms in the CMAB setting. Under the $(\alpha, \beta)$-approximation oracle, a super arm $S$ is considered *sub-optimal* if $r_{\mu}(S) < \alpha \cdot \text{opt}_{\mu}$. Let $\mathcal{S}_{i,B}$ denote the set of all sub-optimal super arms that include base arm $i$. We sort the elements in $\mathcal{S}_{i,B}$ as $S^1_{i,B}, S^2_{i,B}, \ldots, S^{K_i}_{i,B}$ in increasing order of their expected rewards, where $K_i$ is the number of such super arms. The regret gap for the $j$-th sub-optimal super arm is defined as $\Delta^{i,j} = \alpha \cdot \text{opt}_{\mu} - r_{\mu}(S^j_{i,B})$.

**Theorem 5.3** *By setting $i_{S,t} = \text{argmin}_{j \in S} T_j(t)$ and $\gamma_S(t) = N(T_{i_{S,t}}(t))^\varepsilon$, with probability lager than $1 - Kn^{-3}$, the regret upper bound of MSMR-CUCB is*

$$\sum_{i \in \llbracket K \rrbracket, \Delta^i_{min} \geq 0} \left( \ell_n(\Delta^{i,K_i})\Delta^{i,K_i} + \int_{\Delta^{i,K_i}}^{\Delta^{i,1}} \ell_n(x)\,dx \right) + 2KMN\zeta(2-\varepsilon)\Delta_{\max}$$
$$+ KMN \max_{k \in \llbracket K_i \rrbracket, i \in \llbracket K \rrbracket} \{(\ell_n(\Delta^{i,k}))^\varepsilon \Delta^{i,k}\}. \tag{8}$$

*where $\ell_n(\Delta^{i,l}) = (6\ln n)/(f^{-1}(\Delta^{i,l}))^2$, $\Delta^i_{\min} = \Delta^{i,K_i}$ and $\Delta_{\max} = \max_{i \in \llbracket K \rrbracket} \Delta^{i,1}$,*

Theorems 5.2 and 5.3 provide the regret bounds of MSMR algorithms, which dynamically depend on the hyperparameter $\varepsilon$. This parameter controls the trade-off between switching cost and marginal regret, affecting the length of the near-optimal exploitation phase. Its value can be chosen initially based on application requirements and known problem parameters, offering flexibility to adapt the algorithm to different scenarios. Let $Reg_1(n)$ and $Reg_2(n)$ denote the regret upper bounds of MSMR in MAB and CMAB settings, respectively, and $Reg_1^{opt}(n)$ and $Reg_2^{opt}(n)$ the bounds of classical algorithms established in prior work (3; 6). Then, the following relationship holds:

$$\lim_{n\to\infty} \frac{Reg_1(n)}{Reg_1^{opt}(n)} = \lim_{n\to\infty} \frac{Reg_2(n)}{Reg_2^{opt}(n)} = 1. \tag{9}$$

This indicates that our algorithm is asymptotically consistent with the classical counterparts, incurring only minimal marginal regret regardless of the value of $\varepsilon$.

**Theorem 5.4** *Setting* $\gamma_i(t) = N(T_{i_t}(t))^\varepsilon$. *If* $\left(2^{\frac{\varepsilon}{1-\varepsilon}}/(N(1-\varepsilon))\right) \leq \ln(n)$, *the switching cost upper bound of MSMR-UCB is*

$$4KMN \cdot \zeta(2-\varepsilon) + 2\sum_{i\neq i_*}\left(\frac{8\ln(n)}{\Delta_i^2}\right)^{1-\varepsilon} \frac{2^{\frac{\varepsilon}{1-\varepsilon}}}{N(1-\varepsilon)} + 2K.$$

**Theorem 5.5** *Setting* $\gamma_S(t) = N(T_{i_{S_t,t}}(t))^\varepsilon$. *If* $\left(2^{\frac{\varepsilon}{1-\varepsilon}}/(N(1-\varepsilon))\right) \leq \ln(n)$, *the switching cost upper bound of MSMR-CUCB is*

$$4KMN \cdot \zeta(2-\varepsilon) + 2\sum_{i=1}^{K}\left(\frac{6\ln(n)}{(f^{-1}(\Delta^{i,K_i}))^2}\right)^{1-\varepsilon} \frac{2^{\frac{\varepsilon}{1-\varepsilon}}}{N(1-\varepsilon)} + 2\sum_{i=1}^{K}K_i.$$

Theorems 5.4 and 5.5 present the switching cost of the MSMR algorithms, which depend dynamically on the choice of the parameter $\varepsilon$, which plays a leading role. The value of $\varepsilon$ can be still determined at the beginning based on specific application requirements and known problem parameters.

**Lemma 5.6** *If* $i_{t_1} \neq \hat{i}_{*,t_1}$ *is selected at round* $t_1$ *and Eq.5 is hold at round* $t_2 = t_1 + \gamma_{i_{t_1}}(t_1)$, *at least one arm* $i \neq \hat{i}_{*,t_2}$ *will be pulled more than 1 phases before the round* $t'$ *where* $\hat{i}_{*,t_2} = i_{t'}$.

**Lemma 5.7** *If* $S_{t_1} \neq \hat{S}_{*,t_1}$ *is selected at round* $t_1$ *and Eq.6 is hold at round* $t_2 = t_1 + \gamma_{S_{t_1}}(t_1)$, *at least one base arm* $i \notin \hat{S}_{*,t_2}$ *will be pulled more than 1 phases before the round* $t'$ *where* $\hat{S}_{*,t_2} = S_{t'}$.

Lemma 5.6 and 5.7 shows that the *predictive selection* technique predicts whether there exists any arm or base arm will be pulled more than one phases before we exploit the empirical optimal (super) arm. Taking this into consideration, We can directly explore the current selected (super) arm to reduce potential switching cost, which also have a large empirical estimation that creates less regret.

**Theorem 5.8** *With the probability larger than* $1 - Kn^{-4}$ *in MAB and* $1 - Kn^{-3}$ *in CMAB, the marginal loss upper bound of MSMR algorithms is* $\mathcal{O}((\log n)^\varepsilon + \lambda(\log n)^{1-\varepsilon})$.

In most cases, $\lambda$ is constant, and setting $\varepsilon = 0.5$ yields the theoretical minimum marginal loss $\mathcal{O}(\lambda\sqrt{\log n})$. When $\lambda$ depends on $n$, the optimal $\varepsilon$ can be derived, e.g., for $\lambda = \sqrt{\log n}$, the optimal choice is $\varepsilon = 2/3$. From a broader perspective, if the parameter $\varepsilon \in [0,1]$, our proposed MSMR framework encompasses two representative baseline algorithms as special cases. Taking the CMAB setting (Algorithm 1) as an example (ignoring the initial concentrated exploration phase, i.e., lines 2–4), we observe the following limiting cases: When $\varepsilon = 0$, $M = 1$, and $N = 1$, the algorithm degenerates into standard CUCB, which incurs high switching cost. When $\varepsilon = 1$, $M = 1$, $N = \kappa$, and $\gamma_{S_t}(t) = \kappa t$, it reduces to Phased-CUCB, which typically suffers from higher regret. The MSMR framework achieves a better balance between regret and switching cost, lying between these extremes.

## 6 NUMERICAL SIMULATIONS

In this section, we present experiments to assess the performance of our algorithms on both synthetic and real-world datasets. Each experiment was conducted over 20 independent trials to ensure reliability, with

$n = 100000$, $N = 1$, $M = 5$ for all bandit settings and $\alpha = 0.95$ , $\beta = 1$ for CMAB. The tests were performed on a macOS system equipped with an Apple M3 Pro processor and 18 GB of RAM. Here, we present only the experiments for the CMAB setting. For results on MAB settings, ablation studies, and real-world datasets, please refer to Appendix I.

## 6.1 EXPERIMENT SETUP

**Data Generation.** We conduct experiments on cascading bandits, a specific instance of CMAB, comparing against algorithms CUCB (6), phased-CUCB(16), B-FTRL(11). The objective is to select $m = 5$ items from a set of $K = 20$ to maximize the reward. We give a very similar reward distribution where $\mu_i = 0.3 + 0.002 \times i$. In each round $t$, a list $S_t = (a_{t,1}, \ldots, a_{t,m}) \subseteq [\![K]\!]$ is randomly selected. The outcome $X_{t,i}$ for each $i \in S_t$ is generated from a Bernoulli distribution with mean $\mu_i$. Given the ranked list $S_t$, if stopping at the $j_t$-th item, the observed outcomes are: $(X_{t,a_1}, \ldots, X_{t,a_k}) = (0, \ldots, 0, 1, x, \ldots, x)$, where the first $j_t - 1$ items are 0, the $j_t$-th item is 1, and the rest are unobserved ($x$). If the list is exhausted, the observed outcomes are: $(X_{t,a_1}, \ldots, X_{t,a_k}) = (0, 0, \ldots, 0)$. The reward is 1 for stopping and 0 for exhausting the list. The reward function can be written as $r(S_t; \boldsymbol{\mu}) = 1 - \prod_{i \in S_t}(1 - \mu_i)$.

## 6.2 EXPERIMENTAL RESULTS

**Regret, Switching Cost and Marginal Loss.** In Figure 1(a), we observe that the regret of MSMR and MSMR-P closely matches that of the standard baseline methods. In contrast, B-FTRL exhibits noticeably higher regret. Figure 1(b) further shows that standard and phased methods suffer from a substantial number of switches, often exceeding several thousand. B-FTRL also incurs a significant number of switches. In comparison, MSMR results in only 432 in the CMAB setting, amounting to merely 2.4% of the switches incurred by CUCB. Moreover, MSMR-P achieves even greater savings, reducing switching to just 1.3% of CUCB—representing a nearly 50% reduction in switching cost compared to MSMR. These results highlight the effectiveness of the *predictive selection* technique. In terms of marginal loss, figure 1(c) further shows that the MSMR framework achieves remarkably low loss compared to the best existing algorithm, significantly outperforming all other methods.

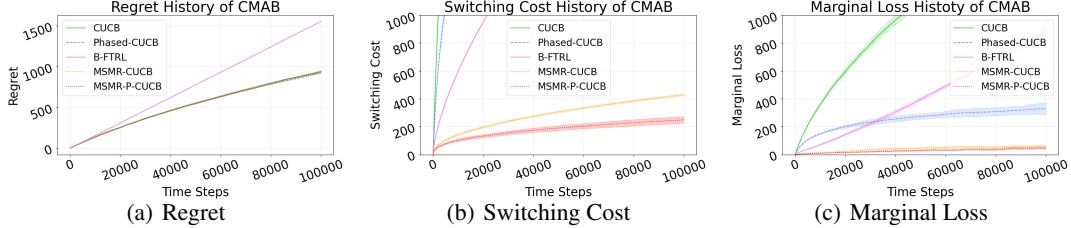

(a) Regret                (b) Switching Cost              (c) Marginal Loss

Figure 1: Synthetic Experiments on CMAB

## 7 CONCLUSION

This paper introduces a novel bandit framework that achieves minimal switching cost and minimal marginal regret, effectively addressing the trade-off between switching costs and regret in bandit algorithms. We develop general techniques—*initial concentrated exploration*, *near-optimal exploitation*, and *predictive selection*, which are broadly applicable to MAB and CMAB settings. Through rigorous theoretical analysis, we establish that these techniques guarantee only $\mathcal{O}((\log n)^{1-\varepsilon})$ switching cost while incurring negligible marginal regret, thereby achieving only $\mathcal{O}(\lambda(\sqrt{\log n})$ marginal loss. Empirical results further demonstrate that MSMR algorithms perform only a few hundred switches, merely 1.3% of those made by standard methods, highlighting the significant advantage of MSMR over existing algorithms. Besides, this paper only provides some theories for part of the bandit scenarios. We hope to extend this framework to more scenarios such as linear or constrained bandit in the future.

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
