# A NOTATION

Table 2: Notation.

| Alphabet | Meanings |
|---|---|
| $c_{i,T_i(t)}$ | confidence interval for arm $i$ in round $t$ |
| $C(n)$ | switching cost |
| $f(\cdot)$ | bounded smoothness function |
| $F_t$ | oracle fails to produce an $\alpha$ approximate answer with respect to input vector $\bar{\mu}_t$ in round $t$ |
| $i$ | arm in MAB or base arm in CMAB |
| $i_{S,t}$ | arm belong to $S$ with lowest pulled time up to round $T$ |
| $i_t$ | arm selected in round $t$ |
| $i_*$ | optimal arm |
| $\hat{i}_*$ | arm with largest empirical mean reward |
| $\mathbb{I}(\cdot)$ | indication function |
| $K$ | number of arms |
| $K_i$ | number of sub-optimal supers arm containing $i$ |
| $K_i'$ | number of sub-optimal super arms containing $i$ indeed pulled in transition stage |
| $L(\cdot)$ | $M \times \gamma(\cdot)$ |
| $L_{i,j}$ | length of $j_{th}$ phase for arm $i$ |
| $\ell(\cdot)$ | function in CUCB(6) |
| $M$ | hyperparameter in MSMR algorithms |
| $n$ | time horizon |
| $N$ | constant in $\gamma(\cdot)$ |
| $N_i$ | counter for base arm $i$ |
| $N_{i,t}$ | counter for base arm $i$ after round $t$ |
| $N_{i,t}^l$ | increasing counter for base arm $i$ due to $\mathcal{S}_{i,B}^l$ after round $t$ |
| $N_{i,t}^{l,suf}$ | increasing counter for base arm $i$ due to $\mathcal{S}_{i,B}^l$ after round $t$ in sufficiently sampled stage |
| $N_{i,t}^{l,und}$ | increasing counter for base arm $i$ due to $\mathcal{S}_{i,B}^l$ after round $t$ in under-sampled stage |
| $\mathcal{N}_t$ | process is nice at round $t$ |
| $\text{opt}_{\boldsymbol{\mu}}$ | expect reward for $S_*$ |
| $r_{\boldsymbol{\mu}}(S)$ | expect reward function for $S$ |
| $R(S)$ | reward function for $S$ |
| $R^A(\lambda, n)$ | marginal loss in time horizon $n$ |
| $Reg(n)$ | regret in time horizon $n$ |
| $Reg_u(n)$ | regret in under-sampled stage in time horizon $n$ |
| $Reg_t(n)$ | regret in transition stage in time horizon $n$ |
| $Reg_s(n)$ | regret in sufficiently sampled stage in time horizon $n$ |
| $S$ | super arm |
| $S_t$ | super arm selected at round $t$ |
| $S_*$ | optimal super arm |
| $\mathcal{S}_{i,B}^j$ | sub-optimal super arms containing arm $i$ with $j_{th}$ lowest expected reward |
| $\hat{S}_{*,t}$ | super arm with largest reward under $r_{\hat{\boldsymbol{\mu}}}(\cdot)$ at round $t$ |
| $\mathcal{S}$ | set of super arms |
| $\mathcal{S}_{i,B}$ | set of all sub-optimal super arms containing arm $i$ |
| $t$ | time round |
| $t_{u,i}$ | number of phases for selecting arm $i$ in under-sampled stage |
| $t_{t,i}$ | number of phases for selecting arm $i$ in transition stage |

Continued on next page

**Table 2 – continued from previous page**

| | |
|---|---|
| $t_{s,i}$ | number of phases for selecting arm $i$ in sufficiently sampled stage |
| $t_s$ | number of phases for selecting any sub-optimal super arm in sufficiently sampled stage |
| $T_i(t)$ | number of times arm $i$ has been pulled up to round $t$ |
| $W_{i,s}$ | sum of $L_{i,1}$ to $L_{i,s}$ |
| $X_{i,t}$ | reward feedback for arm $i$ in round $t$ |
| $\boldsymbol{y}_i$ | item vector for item $i$ |
| $\alpha$ | parameter in $(\alpha, \beta)$ -approximation oracle |
| $\beta$ | parameter in $(\alpha, \beta)$ -approximation oracle |
| $\delta_{i,k}$ | regret gap of $k_{th}$ sub-optimal super arm containing $i$ indeed pulled in transition stage |
| $\Delta_i$ | regret gap for arm $i$ |
| $\Delta^{i,j}$ | regret gap for $S_{i,B}^j$ |
| $\Delta_{\min}^i$ | sub-optimal super arm with smallest regret gap containing $i$ |
| $\Delta_{\max}^i$ | sub-optimal super arm with largest regret gap containing $i$ |
| $\Delta_{\max}$ | sub-optimal super arm with largest regret gap |
| $\varepsilon$ | index in $\gamma(\cdot)$ |
| $\boldsymbol{\theta}_u$ | user preference vector for user $u$ |
| $\gamma(\cdot)$ | exploitation function |
| $\Lambda_{i,t}$ | confidence interval for base arm $i$ in round $t$ in CMAB |
| $\Lambda_t$ | largest confidence interval among all base arm in round $t$ in CMAB |
| $\Lambda^{i,l}$ | $\sqrt{3\ln(n)/2\ell(\Delta^{i,l})}$ |
| $\mu_i$ | expected reward for arm $i$ |
| $\mu_*$ | expected reward for arm $i_*$ |
| $\hat{\mu}_{i,t}$ | empirical mean reward for arm $i$ in round $t$ |
| $\bar{\mu}_{i,t}$ | upper confidence estimate for arm $i$ in round |
| $\bar{\mu}_{i,t}'$ | $\hat{\mu}_{i,t} + \sqrt{2\ln(n)/T_i(t)}$ (MAB) or $\hat{\mu}_{i,t} + \sqrt{3\ln(n)/2T_i(t)}$ (CMAB) |
| $\boldsymbol{\mu}$ | expected reward vector |
| $\hat{\boldsymbol{\mu}}$ | empirical mean reward vector |
| $\bar{\boldsymbol{\mu}}$ | upper confidence estimate vector |
| $\tau_i$ | the start round of $i_{th}$ phase in sufficiently sampled stage of CUCB |
| $\zeta(\cdot)$ | Riemann Zeta Function |

# B PRELIMINARY KNOWLEDGE AND NOTATIONS

We use $[\![K]\!]$ to denote the set $\{1, 2, \ldots, K\}$ for $\forall K \in \mathbb{N}^+$.

**Fact B.1** *(12) (Chernoff-Hoeffding bound). Let $X_1, \cdots, X_n$ be random variables with common support $[0, 1]$ and $\mathbb{E}[X_i] = \mu$. Let $S_n = X_1 + \cdots + X_n$. Then for all $t \geq 0$,*

$$P[S_n \geq n\mu + t] \leq e^{-2t^2/n}, P[S_n \leq n\mu - t] \leq e^{-2t^2/n} \tag{10}$$

# C SUPPLEMENTARY ALGORITHM

---
**Algorithm 2** MSMR-UCB Algorithm
---
**Input:** Time horizon $n$, constant $M$, $\alpha$, function $\gamma(\cdot)$
1: $t \leftarrow 1$, $\hat{\mu}_i \leftarrow 1$ for all $i$
2: **for** all arm $i \in [\![K]\!]$ **do**
3:     Play $i$ and observe $X_{i,t}$ , update $T_i(t) \leftarrow T_i(t) + 1$, $\hat{\mu}_{i,t} \leftarrow \frac{T_i(t-1) \cdot \hat{\mu}_{i,t-1} + X_{i,t}}{T_i(t)}$, $\bar{\mu}_{i,t} \leftarrow \hat{\mu}_{i,t} + \sqrt{2 \ln t / T_i(t)}$ , $\bar{\mu}'_{i,t} \leftarrow \hat{\mu}_{i,t} + \sqrt{2 \ln n / T_i(t)}$ until $\bar{\mu}'_{i,t} \leq 1$ and update $t$
4: **end for**
5: **while** $t \leq n$ **do**
6:     $i_t \leftarrow \operatorname{argmax}_{j \in [\![K]\!]} \bar{\mu}_j$
7:     $Z \leftarrow \gamma_{i_t}(t)$
8:     $\hat{i}_{*,t} \leftarrow \operatorname{argmax}_{j \in [\![K]\!]} \hat{\mu}_j$
9:     $Z \leftarrow M \cdot \gamma_{i_t}(t)$ when $\hat{i}_{*,t} = i_t$
10:     Play arm $i_t \min\{Z, n - t\}$ times
11:     Update $t$ and $T_i(t)$, $\hat{\mu}_{i,t}$, $\bar{\mu}_{i,t}$ for all arms
12: **end while**
---

# D PROOF OF COROLLARY 5.1

## D.1 MAB CASE

By Lemma B.1, if $T_i(t) > l_i$, we have $P(\hat{\mu}_{i,t} - \mu_i - \Delta_i/2 > 0) \leq exp(-2l_i(\Delta_i/2)^2) \leq n^{-4}$, and the following inequality:

---

**Algorithm 3** MSMR-P-UCB Algorithm

---

**Input:** Time horizon $n$, constant $M$, $\alpha$, function $\gamma(\cdot)$

1: $t \leftarrow 1$, $\hat{\mu}_i \leftarrow 1$ for all $i$

2: **for** all arm $i \in [\![K]\!]$ **do**

3:     Play $i$ and observe $X_{i,t}$ , update $T_i(t) \leftarrow T_i(t) + 1$, $\hat{\mu}_{i,t} \leftarrow \frac{T_i(t-1)\cdot\hat{\mu}_{i,t-1}+X_{i,t}}{T_i(t)}$, $\bar{\mu}_{i,t} \leftarrow \hat{\mu}_{i,t} + \sqrt{2\ln t/T_i(t)}$ , $\bar{\mu}'_{i,t} \leftarrow \hat{\mu}_{i,t} + \sqrt{2\ln n/T_i(t)}$ until $\bar{\mu}'_{i,t} \leq 1$ and update $t$

4: **end for**

5: **while** $t \leq n$ **do**

6:     $i_t \leftarrow \operatorname{argmax}_{j\in[\![K]\!]} \bar{\mu}_j$

7:     $Z \leftarrow \gamma_{i_t}(t)$

8:     $\hat{i}_{*,t} \leftarrow \operatorname{argmax}_{j\in[\![K]\!]} \hat{\mu}_j$

9:     $Z \leftarrow M \cdot \gamma_{i_t}(t)$ when $\hat{i}_{*,t} = i_t$

10:    Play arm $i_t \min\{Z, n-t\}$ times

11:    Update $t$ and $T_i(t)$, $\hat{\mu}_{i,t}$, $\bar{\mu}_{i,t}$ for all arms

12:    $\hat{i}_{*,t} \leftarrow \operatorname{argmax}_{j\in[\![K]\!]} \hat{\mu}_j$

13:    $t_3 \leftarrow t + \sum_{j\neq i_t,\hat{i}_{*,t}} \gamma_j(t)$

14:    **if** $\sqrt{2\ln(t)/T_{i_t}(t)} + \hat{\mu}_{i_t,t} \geq \sqrt{2\ln(t_3)/T_{\hat{i}_{*,t}}(t)} + \hat{\mu}_{\hat{i}_{*,t},t}$ **then**

15:       **goto** Line 10

16:    **end if**

17: **end while**

---

$$P(\exists i : T_i(t) \geq l_i, \bar{\mu}'_{i,t} \geq 1) \leq \bigcup_{i \neq i_*} P(T_i(t) \geq l_i, \hat{\mu}_{i,t} + \sqrt{2\ln n/T_i(t)} \geq 1) \tag{11}$$

$$\leq \bigcup_{i \neq i_*} P(\hat{\mu}_{i,t} + \sqrt{2\ln n/l_i} \geq 1) \tag{12}$$

$$= \bigcup_{i \neq i_*} P(\hat{\mu}_{i,t} + \Delta_i/2 \geq 1) \tag{13}$$

$$= \bigcup_{i \neq i_*} P(\hat{\mu}_{i,t} - \mu_i - \Delta_i/2 \geq 1 - \mu_*) \tag{14}$$

$$\leq \bigcup_{i \neq i_*} P(\hat{\mu}_{i,t} - \mu_i - \Delta_i/2 \geq 0) \tag{15}$$

$$\leq Kn^{-4} \tag{16}$$

which means with a probability larger than $1 - Kn^{-4}$, initial concentrated exploration will occur in the under-sampled stage, which is calculated in the total regret.

## D.2   CMAB CASE

By Lemma B.1, if $T_i(t) > \ell_n(\Delta^{i,1})$, we have $P(\hat{\mu}_{i,t} - \mu_i - f^{-1}(\Delta^{i,1})/2 > 0) \leq exp(-2\ell_n(\Delta^{i,1})(f^{-1}(\Delta^{i,1})/2)^2) \leq n^{-3}$, and the following inequality:

---

**Algorithm 4** MSMR-CUCB Algorithm

---

**Input:** Time horizon $n$, constant $M$, $\alpha$, function $\gamma(\cdot)$

1: $t \leftarrow 1$, $\hat{\mu}_i \leftarrow 1$ for all $i$
2: **while** Exists $i$ makes $T_i(t) = 0$ **do**
3: $\quad \{\hat{\mu}_{j_1}, \hat{\mu}_{j_2}, \ldots, \hat{\mu}_{j_K}\} \leftarrow$ Sort base arm by $\hat{\mu}_i$ in a decreasing way
4: $\quad S \leftarrow \{j_1, j_2, \ldots, j_m\}$
5: $\quad$ Play $S$ and observe $X_{i,t}$, for any $i \in S$, update $T_i(t) \leftarrow T_i(t) + 1$, $\hat{\mu}_{i,t} \leftarrow \frac{T_i(t-1)\cdot\hat{\mu}_{i,t-1}+X_{i,t}}{T_i(t)}$,
$\quad \bar{\mu}_{i,t} \leftarrow \hat{\mu}_{i,t} + \sqrt{3\ln t/2T_i(t)}$, $\bar{\mu}'_{i,t} \leftarrow \hat{\mu}_{i,t} + \sqrt{3\ln n/2T_i(t)}$ until $\min_{j\in S} \bar{\mu}'_{j,t} \leq 1$ and update $t$
6: **end while**
7: **while** $t \leq n$ **do**
8: $\quad S_t \leftarrow \arg\max_{S\in\mathcal{S}} r_{\bar{\boldsymbol{\mu}}}(S)$, $Z \leftarrow \gamma_{S_t}(t)$
9: $\quad \hat{S}_{*,t} \leftarrow \arg\max_{S\in\mathcal{S}} r_{\hat{\boldsymbol{\mu}}}(S)$, $Z \leftarrow M \cdot \gamma_{S_t}(t)$ when $r_{\hat{\boldsymbol{\mu}}}(S_t) \geq \alpha \cdot r_{\hat{\boldsymbol{\mu}}}(\hat{S}_{*,t})$
10: $\quad$ Play super arm $S_t \min\{Z, n-t\}$times
11: $\quad$ Update $t$ and $T_i(t)$, $\hat{\mu}_{i,t}$, $\bar{\mu}_{i,t}$ for all base arms
12: $\quad \hat{S}_{*,t} \leftarrow \arg\max_{S\in\mathcal{S}} r_{\hat{\boldsymbol{\mu}}}(S)$
13: $\quad t_3 \leftarrow t + \sum_{j\in\{\llbracket K\rrbracket\setminus(S_t\bigcup\hat{S}_{*,t})\}} \gamma_j(t)$
14: $\quad$ **if** $r(\hat{\boldsymbol{\mu}}_t, \boldsymbol{c}_t, S_t) \geq r(\hat{\boldsymbol{\mu}}_t, \boldsymbol{c}_{t_3}, \hat{S}_{*,t})$ **and** $t < n$ **then**
15: $\quad\quad$ **goto** Line 12
16: $\quad$ **end if**
17: **end while**

---

$$P(\exists i : T_i(t) \geq \ell_n(\Delta^{i,1}), \bar{\mu}'_{i,t} \geq 1) \leq \bigcup_{i\in\llbracket K\rrbracket} P(T_i(t) \geq \ell_n(\Delta^{i,1}), \hat{\mu}_{i,t} + \sqrt{3\ln n/2T_i(t)} \geq 1) \tag{17}$$

$$\leq \bigcup_{i\in\llbracket K\rrbracket} P(\hat{\mu}_{i,t} + \sqrt{3\ln n/2\ell_n(\Delta^{i,1})} \geq 1) \tag{18}$$

$$= \bigcup_{i\in\llbracket K\rrbracket} P(\hat{\mu}_{i,t} + f^{-1}(\Delta^{i,1})/2 \geq 1) \tag{19}$$

$$= \bigcup_{i\in\llbracket K\rrbracket} P(\hat{\mu}_{i,t} - \mu_i - f^{-1}(\Delta^{i,1})/2 \geq 1 - \mu_i - f^{-1}(\Delta^{i,1})) \tag{20}$$

$$\leq \bigcup_{i\in\llbracket K\rrbracket} P(\hat{\mu}_{i,t} - \mu_i - f^{-1}(\Delta^{i,1})/2 \geq 1 - \mu_{\arg\max_{j\in S_*}|\mu_j-\mu_i|}) \tag{21}$$

$$\leq \bigcup_{i\in\llbracket K\rrbracket} P(\hat{\mu}_{i,t} - \mu_i - f^{-1}(\Delta^{i,1})/2 \geq 0) \tag{22}$$

$$\leq \bigcup_{i\neq i_*} P(\hat{\mu}_{i,t} - \mu_i - \Delta_i/2 \geq 0) \tag{23}$$

$$\leq Kn^{-3} \tag{24}$$

which means with a probability larger than $1 - Kn^{-3}$, initial concentrated exploration will occur in the under-sampled stage, which is calculated in the total regret.

# E   PROOF OF REGRET

## E.1   PROOF OF THEOREM 5.2

Let $[\![K]\!]$ denotes the set of arms. For each arm $i \in [\![K]\!]$, pulling it at round $t$ yields a reward feedback $X_{i_t,t} \in [0, 1]$. The unknown mean vector is represented by $\boldsymbol{\mu} = (\mu_1, ..., \mu_K)$, where $\mu_i = \mathbb{E}[X_{i,t}]$ denotes the expected reward for any base arm $i$. Besides, we define $\Delta_i = \mu_* - \mu_i$. The arm with the highest expected reward is called the optimal arm, and its mean reward is denoted by $\mu_* = \max_{i \in [\![K]\!]} \mu_i$. The objective of the MAB problem is to identify this optimal arm while minimizing regret, which is defined as:

$$Reg(n) = n\mu_* - \mathbb{E}[\sum_{t=1}^{n} X_{i_t,t}], \tag{25}$$

where $n$ is the total time horizon, and $X_{i_t,t}$ represents the reward from the chosen arm at round $t$.

In MAB settings, we set that $\gamma_i(t) = N(T_{i_t}(t))^\varepsilon$ is the number of pulling times for the selected arm $i_t$, where $N$ is a constant, $0 < \varepsilon < 1$ is a hyperparameter. If the selected arm $i_t = \operatorname{argmax}_{j \in [\![K]\!]} \bar{\mu}_{j,t}$ has the largest upper confidence estimation, which is $\hat{i}_* = \operatorname{argmax}_{j \in [\![K]\!]} \hat{\mu}_{j,t}$ and $i_t = \hat{i}_*$, agent assume that arm $i_t$ has the relatively large empirical estimation. Under this condition, we execute the near-optimal exploitation and set $M \cdot \gamma_i(t) = MN(T_{i_t}(t))^\varepsilon$ as the exploitation times, where $M$ is a constant. We decompose the regret of MSMR-UCB as three stages: (1) under-sampled stages; (2) transition stages; (3) sufficiently sampled stages.

**Under-sampled stages.** This stage is a fixed regret for a certain arm in expectation. According to (15), we know that each arm $i$ must have some inevitable exploring regret in UCB-based methods, which is $\mathcal{O}(\log n)$. In MSMR-UCB, we set under-sampled regret number of arm $i$ is $l_i = 8\ln(n)/\Delta_i^2$ and arm $i$ is under-sampled or sufficiently sampled if $T_i(t) \leq l_i$ or $T_i(t) \geq l_i$. By our definition, the regret of under-sampled stage is at most:

$$Reg_u(n) = \sum_{i \neq i_*} l_i \Delta_i \leq \sum_{i \neq i_*} \frac{8\ln n}{\Delta_i}. \tag{26}$$

**Transition stages.** When transiting from under-sampled stages to the sufficiently sampled cases, we may pull some arm $i$ more times. By the definition of and near-optimal exploitation, we at most pull arm $i$ at most $M\gamma_i(t)$ more times. The regret of transition stages is at most:

$$Reg_t(n) \leq \sum_{i \neq i_*} M\gamma_i(t)\Delta_i \tag{27}$$

$$\leq \sum_{i \neq i_*} MN(l_i)^\varepsilon \Delta_i \tag{28}$$

$$\leq \sum_{i \neq i_*} MN(8\ln n)^\varepsilon \Delta_i^{1-2\varepsilon}. \tag{29}$$

**Sufficiently sampled stages.** We define arm $i$ as sufficiently sampled if $T_i(t) \geq l_i$. At sufficiently sampled stages, each suboptimal arm has been pulled enough times. In this situation, we have to get much information for any suboptimal arm $i$, it has a low probability to choose a suboptimal arm $i$. Specifically, denoting $\bar{\mu}_{i,T_i(t)}$

is the upper confidence estimate of arm $t$ in round $t$, we have the following inequality:

$$\mathbb{E}[T_i(n)] = \sum_{t=l+1}^{n} P(\bar{\mu}_{i,T_i(t)} = \max_{j \in [\![K]\!]} \bar{\mu}_{j,T_j(t)}, T_i(t) \geq l)(\gamma_i(t) + (M-1)\gamma_i(t)P(i = \hat{i}_*)) \tag{30}$$

$$\leq \sum_{t=l+1}^{n} P(\bar{\mu}_{i,T_i(t)} = \max_{j \in [\![K]\!]} \bar{\mu}_{j,T_j(t)}, T_i(t) \geq l) \cdot M\gamma_i(t) \tag{31}$$

$$\leq \sum_{t=l+1}^{n} P(\bar{\mu}_{*,T_*(t)} \leq \bar{\mu}_{i,T_i(t)}, T_i(t) \geq l) \cdot M\gamma_i(t) \tag{32}$$

$$\leq \sum_{t=l+1}^{n} P(\min_{1<s<t} \bar{\mu}_{*,T_*(t)} \leq \max_{l \leq s_i \leq t} \bar{\mu}_{i,T_i(t)}) \cdot M\gamma_i(t) \tag{33}$$

$$\leq \sum_{t=1}^{\infty} \sum_{s=1}^{t-1} \sum_{s_i=l}^{t-1} P(\bar{\mu}_{*,s} \leq \bar{\mu}_{i,s_i}) \cdot M\gamma_i(t) \tag{34}$$

$$\leq \sum_{t=1}^{\infty} \sum_{s=1}^{t-1} \sum_{s_i=l}^{t-1} (P(\hat{\mu}_{*,s} \leq \mu_* - c_{*,s}) + P(\mu_i + c_{i,s_i} \leq \hat{\mu}_{i,s_i})) \cdot M\gamma_i(t) \tag{35}$$

$$\leq \sum_{t=1}^{\infty} \sum_{s=1}^{t-1} \sum_{s_i=l}^{t-1} \frac{2}{t^4} \cdot M\gamma_i(t) \tag{36}$$

$$\leq \sum_{t=1}^{\infty} \frac{2}{t^2} \cdot M\gamma_i(t) \tag{37}$$

$$\leq \sum_{t=1}^{\infty} \frac{2}{t^2} MNt^{\varepsilon} \tag{38}$$

$$= 2MN \cdot \zeta(2 - \varepsilon) \tag{39}$$

$$\tag{40}$$

So, the regret of sufficiently sampled stage is:

$$Reg_s(n) = \sum_{i \neq i_*} 2MN \cdot \zeta(2 - \varepsilon)\Delta_i \tag{41}$$

In summary, the regret upper bound is:

$$Reg(n) = Reg_u(n) + Reg_t(n) + Reg_s(n) \tag{42}$$

$$\leq \sum_{i \neq i_*} \left( \frac{8 \ln n}{\Delta_i} + MN(8 \ln(n))^{\varepsilon} \Delta_i^{1-2\varepsilon} + 2MN \cdot \zeta(2 - \varepsilon)\Delta_i \right). \tag{43}$$

## E.2  PROOF OF THEOREM 5.3

In CMAB settings, we choose the super arm $S_t$ who has the largest upper confidence estimation, which is $S_t = \text{argmax}_{S \in \mathcal{S}} r_{\bar{\mu}}(S)$. We set that $\gamma_S(t) = N(T_{i_{S_t,t}}(t))^{\varepsilon}$ is the number of pulling times for the selected super arm $S_t$, where $M$ is a constant, $0 < \varepsilon < 1$ is a hyperparameter and $i_{S,t} = \text{argmin}_{j \in S} T_j(t)$. Based

on $(\alpha, \beta)$-approximation and defining $\hat{S}_* = \arg\max_{S \in \mathcal{S}} r_{\hat{\mu}}(S)$, if $r_{\hat{\mu}}(S_t) \geq \alpha \cdot r_{\hat{\mu}}(\hat{S}_*)$, we think $S_t$ has relatively large empirical estimation. Under this condition, we execute the near-optimal exploitation and set $M \cdot \gamma(t) = MN(T_{i_{S_t,t}}(t))^\varepsilon$ as the exploitation times, where $N$ is a constant.

In the $(\alpha, \beta)$-approximation oracle setting, a super arm $S$ is considered *sub-optimal* if $r_{\mu}(S) < \alpha \cdot \mathrm{opt}_{\mu}$. Let $\mathcal{S}_{i,B}$ be the set of all sub-optimal super arms containing base arm $i$. We sort all sub-optimal super arms in $\mathcal{S}_{i,B}$ as $S_{i,\mathrm{B}}^1, S_{i,\mathrm{B}}^2, \ldots, S_{i,\mathrm{B}}^{K_i}$, in increasing order of their expected rewards, where $K_i$ the number of sub-optimal super arms containing base arm $i$. Define the regret gap for the $j$-th sub-optimal super arm as: $\Delta^{i,j} = \alpha \cdot \mathrm{opt}_{\mu} - r_{\mu}(S_{i,\mathrm{B}}^j)$ and denote $\Delta_{\max}^i = \Delta^{i,1}$, $\Delta_{\min}^i = \Delta^{i,K_i}$ and $\Delta_{\max} = \max_{i \in [\![K]\!]} \Delta_{\max}^i$. And for each $S_{i,\mathrm{B}}^l$, we define sufficient sampling of $i$ with respect to $S_{i,\mathrm{B}}^l$ as i being sampled $\ell_n(\Delta^{i,l}) = (6 \ln n)/(f^{-1}(\Delta^{i,l}))^2$ times.

For the proof, we maintain counter $N_i$ for each arm $i$. Let $N_{i,t}$ be the value of $N_i$ after the $t$-th round and $N_{i,0} = 0$. Counters $\{N_i\}_{i=1}^K$ are updated in the following way: For a round $t > K$, let $S_t$ be the super arm selected in round $t$ by the MSMR-CUCB. Round $t$ is a bad round if the oracle selects a super arm $S_t \in \mathcal{S}_{\mathrm{B}}$, which is not an $\alpha$-approximate super arm with respect to the true mean vector $\mu$. If $S_t$ is chosen and round $t$ is bad, we increment $N_{i_{S_t,t}}$ by one, i.e., $N_{i_{S_t,t},t} = N_{i_{S_t,t},t-1} + 1$. In other words, we find the arm $i_{S_t,t}$ with the smallest counter in $S_t$ and increase its counter. If $i_{S_t,t}$ is not unique, we pick an arbitrary arm with the smallest counters in $S_t$. By definition, we know $N_{i_{S_t,t},t} \leq T_i(t)$. Notice that in every bad round, exactly one counter in $\{N_i\}_{i=1}^K$ is increased. Each time $N_i$ gets updated, one of the sub-optimal super arms containing $i$ is played. We further divide counter $N_i$ into more counters $\{N_i^l\}_{l=1}^{K_i}$, whose value at round $n$, $N_{i,n}^l$ is defined as follows:

$$\forall l \in [\![K_i]\!], N_{i,n}^l = \sum_{t=K+1}^{n} \mathbb{I}\{S_t = S_{i,\mathrm{B}}^l, N_{i,t} > N_{i,t-1}\} \tag{44}$$

Notice that every arm in $S_t$ must have been played at least $N_{i,t-1}$ times by round $t$, since in our updating rule we choose the smallest counter value among arms in $S_t$ to update, and $i$ is the chosen one. If $N_{i,t-1} > \ell_n(\Delta^{i,l})$, we say that the bad arm $S_{i,\mathrm{B}}^l$ is sufficiently sampled. Otherwise, it is under-sampled. For our proof, we depart the $N_{i,n}^l$ into two parts: sufficiently sampled parts $N_{i,n}^{l,suf}$ and under-sampled parts $N_{i,n}^{l,Sund}$.

$$N_{i,n}^{l,suf} = \sum_{t=K+1}^{n} \mathbb{I}\{S_t = S_{i,\mathrm{B}}^l, N_{i,t} > N_{i,t-1}, N_{i,t-1} > \ell_n(\Delta^{i,l})\} \tag{45}$$

$$N_{i,n}^{l,und} = \sum_{t=K+1}^{n} \mathbb{I}\{S_t = S_{i,\mathrm{B}}^l, N_{i,t} > N_{i,t-1}, N_{i,t-1} \leq \ell_n(\Delta^{i,l})\} \tag{46}$$

Then we have $N_{i,n} = \sum_{l \in [K_i]}(N_{i,n}^{l,suf} + N_{i,n}^{l,und})$. Using this notation, the total reward at time horizon $n$ is at least

$$n \cdot \alpha \cdot \mathrm{opt}_{\mu} - \mathbb{E}\left[\sum_{i \in [\![K]\!]}\left(\sum_{l \in [\![K_i]\!]}(N_{i,n}^{l,suf} + N_{i,n}^{l,und}) \cdot \Delta^{i,l}\right)\right], \tag{47}$$

where $\Delta^{i,1}$ is for the initialization. By the regret definition of $(\alpha, \beta)$-approximation oracle, the regret of MSMR-CUCB can be written as:

$$(\beta - 1) \cdot n \cdot \alpha \cdot \mathrm{opt}_{\mu} + \mathbb{E}\left[\sum_{i \in [\![K]\!]}\left(\sum_{l \in [\![K_i]\!]}(N_{i,n}^{l,suf} + N_{i,n}^{l,und}) \cdot \Delta^{i,l}\right)\right]. \tag{48}$$

We decompose the regret of MSMR-CUCB as three stages: (1) under-sampled stages; (2) transition stages; (3) sufficiently sampled stages.

**Under-sampled stages.** In MSMR-CUCB, we set under-sampled regret number of super arm $S_{i,\mathrm{B}}^l$ is $\ell_n(\Delta^{i,l})$ and $S_{i,\mathrm{B}}^l$ is under-sampled if $N_{i,t-1} \leq \ell_n(\Delta^{i,l})$. For a specific arm $i$, its counter $N_i$ will increase from 1 to $\ell_n(\Delta^{i,K_i})$. To simplify the notation, let $\ell_n(\Delta^{i,0}) = 0$. (Note that $N_{i,K} = 1$ for all $i$.) Before diving into the details, we briefly explain the essential idea behind Eq. (58). The range of the counter $N_i$ is divided into discrete segments, i.e., $(\ell_n(\Delta^{i,j-1}), \ell_n(\Delta^{i,j})]$ for $j \in [\![K_i]\!]$. Suppose that round $t$ is bad and $N_{i,t-1} \in (\ell_n(\Delta^{i,j-1}), \ell_n(\Delta^{i,j})]$ for some $j$. Note that we are only interested in cases where $S_t$ is under-sampled. Specifically, this means $S_t = S_B^l$ for some $l > j$. (Otherwise, if $S_t$ is sufficiently sampled, it is based on the counter value $N_{i,t-1}$, and no regret is incurred.)

Consequently, for counter $N_{i,t}$ in the range $(\ell_n(\Delta^{i,j-1}), \ell_n(\Delta^{i,j})]$, the bad super arm will suffer a regret of $\Delta^{i,j}$ each time $N_i$ is incremented, as indicated by Eq. (52). Therefore, the total regret incurred for these under-sampled arms within this interval is at most $(\ell_n(\Delta^{i,j}) - \ell_n(\Delta^{i,j-1})) \cdot \Delta^{i,j}$ (refer to Eq. (56)). We formalize this argument as follows. For any arm $i \in \{i \in [\![K]\!] \mid \Delta_{\min}^i > 0\}$, we have:

$$Reg_u(n) = \sum_{l \in [\![K_i]\!]} N_{i,n}^{l,und} \cdot \Delta^{i,l} \tag{49}$$

$$= \sum_{t=K+1}^{} \sum_{l \in [\![K_i]\!]} \mathbb{I}\{S_t = S_{i,\mathrm{B}}^l, N_{i,t} > N_{i,t-1}, N_{i,t-1} \leq \ell_n(\Delta^{i,l})\} \cdot \Delta^{i,l} \tag{50}$$

$$= \sum_{t=K+1}^{n} \sum_{l \in [\![K_i]\!]} \sum_{j=1}^{l} \mathbb{I}\{S_t = S_{i,\mathrm{B}}^l, N_{i,t} > N_{i,t-1}, N_{i,t-1} \in (\ell_n(\Delta^{i,j-1}), \ell_n(\Delta^{i,j})]\} \cdot \Delta^{i,l} \tag{51}$$

$$\leq \sum_{t=K+1}^{n} \sum_{l \in [\![K_i]\!]} \sum_{j=1}^{l} \mathbb{I}\{S_t = S_{i,\mathrm{B}}^l, N_{i,t} > N_{i,t-1}, N_{i,t-1} \in (\ell_n(\Delta^{i,j-1}), \ell_n(\Delta^{i,j})]\} \cdot \Delta^{i,\boldsymbol{j}} \tag{52}$$

$$\leq \sum_{t=K+1}^{n} \sum_{l \in [\![K_i]\!]} \sum_{j \in [\![K_i]\!]} \mathbb{I}\{S_t = S_{i,\mathrm{B}}^l, N_{i,t} > N_{i,t-1}, N_{i,t-1} \in (\ell_n(\Delta^{i,j-1}), \ell_n(\Delta^{i,j})]\} \cdot \Delta^{i,j}$$
$$\tag{53}$$

$$= \sum_{t=K+1}^{n} \sum_{j \in [\![K_i]\!]} \mathbb{I}\{S_t \in \mathcal{S}_{i,\mathrm{B}}, N_{i,t} > N_{i,t-1}, N_{i,t-1} \in (\ell_n(\Delta^{i,j-1}), \ell_n(\Delta^{i,j})]\} \cdot \Delta^{i,j} \tag{54}$$

$$= \sum_{j \in [\![K_i]\!]} \sum_{t=K+1}^{n} \mathbb{I}\{S_t \in \mathcal{S}_{i,\mathrm{B}}, N_{i,t} > N_{i,t-1}, N_{i,t-1} \in (\ell_n(\Delta^{i,j-1}), \ell_n(\Delta^{i,j})]\} \cdot \Delta^{i,j} \tag{55}$$

$$\leq \sum_{j \in [\![K_i]\!]} (\ell_n(\Delta^{i,j}) - \ell_n(\Delta^{i,j-1})) \cdot \Delta^{i,j} \tag{56}$$

$$= \ell_n(\Delta^{i,K_i}) \Delta^{i,K_i} + \sum_{j=1}^{K_i-1} \ell_n(\Delta^{i,j}) \cdot (\Delta^{i,j} - \Delta^{i,j+1}) \tag{57}$$

$$\leq \ell_n(\Delta^{i,K_i}) \Delta^{i,K_i} + \int_{\Delta^{i,K_i}}^{\Delta^{i,1}} \ell_n(x) \mathrm{d}x. \tag{58}$$

Naturally, if we follow the design of MSMR-UCB, each super arm $S$ is pulled an additional $N(T_S(t))^\varepsilon$ times in Algorithm 1, where $T_S(t)$ denotes the number of times $S$ has been selected up to round $t$. Maintaining $T_S(t)$ for all possible super arms would require tracking up to $2^K$ different values, which leads to a combinatorial explosion in both computation and storage. To address this issue, we revise the definition of a *sufficiently sampled* super arm to facilitate the analysis of regret during different sampling stages. Specifically, we say that a sub-optimal super arm $S_{i,\mathrm{B}}^l$ is sufficiently sampled if $T_{i,t-1} > \ell_n(\Delta^{i,l})$. This new definition, which depends only on the sampling counts of individual base arms, is adopted throughout the subsequent analysis. As a result, the algorithm only needs to maintain $T_i(t)$ for each base arm $i$, thereby avoiding the combinatorial overhead. We now discuss the implications of this modified definition. Under the original definition, the sufficiently sampled component $N_{i,n}^{l,\mathrm{suf}}$ and the under-sampled component $N_{i,n}^{l,\mathrm{und}}$ were disjoint. In contrast, with the new definition, these two components may partially overlap. However, this overlap does not affect the correctness of the regret decomposition: all sources of regret are still accounted for. Furthermore, the regret incurred in the overlapping region is negligible, as the primary contribution to the total regret originates from the under-sampled phase.

**Transition stages.** When entering the sufficiently sampled phase, a super arm $S$ may be pulled additional times. Under the revised definition of a sufficiently sampled arm, we consider two possible criteria to determine the round marking the transition stage for a base arm $i$:

- The round $t$ such that $N_{i,t} > \ell_n(\Delta^{i,l})$ and $N_{i,t-1} \leq \ell_n(\Delta^{i,l})$, indicating that the arm has just exited the under-sampled phase.

- The round $t$ such that $T_i(t) > \ell_n(\Delta^{i,l})$ and $T_i(t-1) \leq \ell_n(\Delta^{i,l})$, indicating that the arm has just entered the sufficiently sampled phase.

We adopt the second criterion to define the transition stage. Given a fixed sub-optimal super arm $S_{i,\mathrm{B}}^l$ and its corresponding base arm $i$, $\ell_n(\Delta^{i,l})$ is fixed. Thus, each such super arm may only be "overpulled" once during the transition stage involving arm $i$. Prior to this transition, at least one base arm $j \in S$ must satisfy $T_j(t) < \ell_n(\Delta^{i,l})$. Since $T_i(t)$ monotonically increases over time, this guarantees progress toward the transition.

For a given base arm $i$, we define $\delta_{i,k}$ as the regret gap of the $k$-th sub-optimal super arm (among those that are actually pulled during the learning process), where $k \in [\![K_i']\!]$ and $K_i'$ is a constant no greater than $K_i$, representing the number of such super arms that undergo transition stages involving arm $i$.

When the $k$-th sub-optimal super arm is pulled, it may incur up to $L(\ell_n(\delta_{i,k})) \triangleq MN(\ell_n(\delta_{i,k}))^\varepsilon$ additional pulls (referred to as "overpulls") during its transition stage. Each overpull contributes at most $\delta_{i,k}$ regret. However, the next regret gap $\delta_{i,k+1}$ has already been accounted for in the under-sampled phase of the $(k+1)$-th sub-optimal super arm. Therefore, the additional regret contribution from the transition stage is bounded by $(\delta_{i,k} - \delta_{i,k+1})L(\ell_n(\delta_{i,k}))$.

The total regret incurred during the transition stage involving base arm $i$ is thus given by:

$$\delta_{i,K_i'}L(\ell_n(\delta_{i,K_i'})) + \sum_{k=1}^{K_i'-1}(\delta_{i,k} - \delta_{i,k+1})L(\ell_n(\delta_{i,k})) \tag{59}$$

$$=\delta_{i,1}L(\ell_n(\delta_{i,1})) + \sum_{k=2}^{K_i'}\delta_{i,k}\left[L(\ell_n(\delta_{i,k})) - L(\ell_n(\delta_{i,k-1}))\right] \tag{60}$$

$$\leq\delta_{i,1}L(\ell_n(\delta_{i,1})) \tag{61}$$

$$\leq \max_{k\in[\![K_i]\!]}L(\ell_n(\Delta^{i,k}))\Delta^{i,k} \tag{62}$$

$$= \max_{k\in[\![K_i]\!]}MN(\ell_n(\Delta_{i,k}))^{\varepsilon}\Delta^{i,k}. \tag{63}$$

Then the regret upper bound for transition stage is:

$$Reg_t(n) = \sum_{i\in[\![K]\!]}\max_{k\in[\![K_i]\!]}\{MN(\ell_n(\Delta_{i,k}))^{\varepsilon}\Delta^{i,k}\} \tag{64}$$

$$= KMN\max_{k\in[\![K_i]\!],i\in[\![K]\!]}\{(\ell_n(\Delta^{i,k}))^{\varepsilon}\Delta^{i,k}\} \tag{65}$$

$$\tag{66}$$

To finish our proof, we first give some definitions and lemmas. The learning process is nice at time horizon $t$ if:

$$\forall i \in [\![K]\!], |\hat{\mu}_{i,T_i(t-1)} - \mu_i| < \sqrt{\frac{3\ln t}{2T_i(t-1)}}. \tag{67}$$

**Lemma E.1** *The probability that the process is nice at round $t$ is at least $1 - 2Kt^{-2}$.*

By Chernoff-Hoeffding bound in Fact B.1, for all $i \in [\![K]\!]$, we have:

$$P\left[|\hat{\mu}_{i,T_i(t-1)} - \mu_i| \geq \sqrt{\frac{3\ln t}{2T_i(t-1)}}\right] \tag{68}$$

$$=\sum_{s=1}^{t-1}\Pr\left[\left\{|\hat{\mu}_{i,s} - \mu_i| \geq \sqrt{\frac{3\ln t}{2s}}, T_i(t-1) = s\right\}\right] \tag{69}$$

$$=\sum_{s=1}^{t-1}\Pr\left[\left\{|\hat{\mu}_{i,s} - \mu_i| \geq \sqrt{\frac{3\ln t}{2s}}\right\}\right] \tag{70}$$

$$=\leq t \cdot 2e^{-3\ln t} = \frac{2}{t^2}. \tag{71}$$

**Lemma E.2** *For any time horizon* $n > t > K$

$$\mathbb{E}\left[\sum_{i\in[\![K]\!],l\in[\![K_i]\!]} \mathbb{I}\{S_t = S_{i,B}^l, N_{i,t} > N_{i,t-1}, T_i(t-1) > \ell_n(\Delta^{i,l})\}\right] \tag{72}$$

$$= \sum_{i\in[\![K]\!],l\in[\![K_i]\!]} \Pr\{S_t = S_{i,B}^l, N_{i,t} > N_{i,t-1}, \forall s \in S_{i,B}^l, T_s(t-1) > \ell_n(\Delta^{i,l})\} \tag{73}$$

$$\leq (1-\beta) + 2Kt^{-2}. \tag{74}$$

Define $\Lambda_{i,t} = \sqrt{\frac{3\ln t}{2T_i(t-1)}}$ (a random variable since $T_i(t-1)$ is a random variable) and $\Lambda_t = \max\{\Lambda_{i,t} \mid i \in S_t\}$. Define $\Lambda^{i,l} = \sqrt{\frac{3\ln t}{2\ell_n(\Delta^{i,l})}}$ (not a random variable).

Let $\mathcal{N}_t$ indicate the event that the process is nice at round $t$. Let $F_t$ be the event that the oracle fails to produce an $\alpha$-approximate answer with respect to input vector $\bar{\mu}_t$ in round $t$. We have $\Pr[F_t] = \mathbb{E}[\mathbb{I}\{F_t\}] \leq 1 - \beta$.

Notice that $\bar{\mu}_{i,t} = \hat{\mu}_{i,t} + \sqrt{\frac{3\ln t}{2T_i(t-1)}}$. We have the following properties.

$$\mathcal{N}_t \Rightarrow \forall i \in [\![K]\!], \bar{\mu}_{i,t} - \mu_i > 0 \tag{75}$$
$$\mathcal{N}_t \Rightarrow \forall i \in S_t, \bar{\mu}_{i,t} - \mu_i < 2\Lambda_t, \tag{76}$$
$$\forall i \in [\![K]\!], \forall l \in [\![K_i]\!], \{S_t = S_{i,B}^l, N_{i,t} > N_{i,t-1}, \forall s \in S_t, T_s(t-1) > \ell_n(\Delta^{i,l})\} \Rightarrow \Lambda^{i,l} > \Lambda_t. \tag{77}$$

For any particular $i \in [\![K]\!]$ and $l \in [\![K_i]\!]$, if $\{\mathcal{N}_t, \neg F_t, S_t = S_{i,B}^l, N_{i,t} > N_{i,t-1}, \forall s \in S_t, T_s(t-1) > \ell_n(\Delta^{i,l})\}$ holds at round $t$, we have the following properties:

$$\begin{aligned}
r_{\boldsymbol{\mu}}(S_t) + f(2\Lambda^{i,l}) &> r_{\boldsymbol{\mu}}(S_t) + f(2\Lambda_t) && \text{strict monotonicity of } f(\cdot) \text{ and Eq. 77}\\
&\geq r_{\bar{\boldsymbol{\mu}}_t}(S_t) && \text{bounded smoothness property and Eq.76}\\
&\geq \alpha \cdot \text{opt}_{\bar{\boldsymbol{\mu}}_t} && \neg F_t \Rightarrow S_t \text{ is an } \alpha \text{ approximation w.r.t } \bar{\boldsymbol{\mu}}_t\\
&\geq \alpha \cdot r_{\bar{\boldsymbol{\mu}}_t}(S_*) && \text{definition of opt}_{\bar{\boldsymbol{\mu}}_t}\\
&\geq \alpha \cdot r_{\boldsymbol{\mu}}(S_*) = \alpha \cdot \text{opt}_{\boldsymbol{\mu}}. && \text{monotonicity of } r_{\boldsymbol{\mu}}(S) \text{ and Eq.75}
\end{aligned}$$

So we have

$$r_{\mu}(S_{i,B}^l) + f(2\Lambda^{i,l}) > \alpha \cdot \text{opt}_{\mu}. \tag{78}$$

Since $\ell_n(\Delta^{i,l}) = \frac{6\ln n}{(f^{-1}(\Delta^{i,l}))^2}$, we have $2\Lambda^{i,l} = f^{-1}(\Delta^{i,l}) \cdot \sqrt{\frac{\ln t}{\ln n}}$ which implies $f(2\Lambda^{i,l}) \leq \Delta^{i,l}$ by the monotonicity of $f(\cdot)$ and $t \leq n$. Therefore, Eq.(23) contradicts the definition of $\Delta^{i,l}$ in Eq.(14). In other words,

$$\forall i \in [\![K]\!], \forall l \in [\![K_i]\!], \Pr\{\{\mathcal{N}_t, \neg F_t, S_t = S_{i,B}^l, N_{i,t} > N_{i,t-1}, \forall s \in S_t, T_s(t-1) > \ell_n(\Delta^{i,l})\} = 0 \tag{79}$$

$$\Rightarrow \Pr\{\mathcal{N}_t, \neg F_t, \exists i \in [\![K]\!], \exists l \in [\![K_i]\!], S_t = S_{i,B}^l, N_{i,t} > N_{i,t-1}, \forall s \in S_t, T_s(t-1) > \ell_n(\Delta^{i,l})\} = 0 \tag{80}$$

$$\Rightarrow \Pr\{\exists i \in [\![K]\!], \exists l \in [\![K_i]\!], S_t = S_{i,B}^l, N_{i,t} > N_{i,t-1}, \forall s \in S_t, T_s(t-1) > \ell_n(\Delta^{i,l})\} \tag{81}$$

$$\leq \Pr[F_t \vee \neg \mathcal{N}_t] \leq (1-\beta) + 2Kt^{-2} \tag{82}$$

$$\Rightarrow \sum_{i\in[\![K]\!],l\in[\![K_i]\!]} \Pr\{S_t = S_{i,B}^l, N_{i,t} > N_{i,t-1}, \forall s \in S_t, T_s(t-1) > \ell_n(\Delta^{i,l})\} \leq (1-\beta) + 2Kt^{-2}. \tag{83}$$

The second inequality in Eq.82 uses the facts that $\Pr\{F_t\} = (1 - \beta)$ and $\Pr\{\neg \mathcal{N}_t\} \leq 2Kt^{-2}$ (Lemma E.1). The left side of Eq.83 equals the left side of Eq.82, because the events $\{S_t = \mathcal{S}_{i,B}^l, N_{i,t} > N_{i,t-1}, \forall s \in S_t, T_s(t-1) > \ell_t(\Delta^{i,l})\}$ for all $i \in [\![K]\!]$ and $l \in [\![K_i]\!]$ are mutually exclusive, which in turn is because in each round when $S_t$ is sub-optimal, only one arm $i \in S_t$ gets to increment its counter $N_i$ and thus $N_{i,t} > N_{i,t-1}$, and within arm $i$, only one index $l$ satisfies $S_t = \mathcal{S}_{i,B}^l$.

**Sufficiently sampled stage.** In this stage, if we choose a sub-optimal super arms to pull and pull it more times from $\tau_i$ to $\tau_{i+1}$, it in fact occupies all the exploring chances from $\tau_i$ to $\tau_{i+1}$, so the regret upper bound of this stage is:

$$\mathbb{E}\left[\sum_{t=\tau_1, \tau_2, \dots} P\left(\sum_{i \in [\![K]\!], l \in [\![K_i]\!]} \mathbb{I}\{S_t = \mathcal{S}_{i,B}^l, N_{i,t} > N_{i,t-1}, T_i(t-1), \ell_n(\Delta^{i,l})\}\right) \times \tag{84}$$

$$\left. (\gamma_{S_t}(t) + (M-1)\gamma_{S_t}(t)\mathbb{I}(C)) \right] \Delta_{max} \tag{85}$$

where $\tau_{i+1} = \tau_i + \gamma_{S_t}(t) + (M-1)\gamma_{S_t}(t)\mathbb{I}(C)$ and $\mathbb{I}(C) = 1$ means we exploit this sub-optimal super arm more times via near-optimal exploitation. $\tau_{i+1} - \tau_i$ is the round span of 'overpull' this sub-optimal super arm.

Then we can derive the regret of sufficiently sampled cases:

$$Reg_s(n) \tag{86}$$

$$\leq \mathbb{E}\left[\sum_{t=\tau_1, \tau_2, \dots} ((1 - \beta) + 2Kt^{-2})(\gamma_{S_t}(t) + (M-1)\gamma_{S_t}(t)\mathbb{I}(C))\right] \Delta_{max} \tag{87}$$

$$= \mathbb{E}\left[\sum_{t=\tau_1, \tau_2, \dots} (1 - \beta)(\gamma_{S_t}(t) + (M-1)\gamma_{S_t}(t)\mathbb{I}(C))\right] \Delta_{max} + \tag{88}$$

$$\mathbb{E}\left[\sum_{t=\tau_1, \tau_2, \dots} 2Kt^{-2}(\gamma_{S_t}(t) + (M-1)\gamma_{S_t}(t)\mathbb{I}(C))\right] \Delta_{max} \tag{89}$$

$$\leq \mathbb{E}\left[\sum_{t=\tau_1, \tau_2, \dots} (1 - \beta)(\gamma_{S_t}(t) + (M-1)\gamma_{S_t}(t)\mathbb{I}(C))\right] \Delta_{max} + \tag{90}$$

$$\mathbb{E}\left[\sum_{t=\tau_1, \tau_2, \dots} 2Kt^{-2}(M\gamma_{S_t}(t))\right] \Delta_{max} \tag{91}$$

$$\leq (1 - \beta)n\Delta_{max} + \sum_{t \geq K} 2KMt^{-2}\gamma_{S_t}(t)\Delta_{max} \tag{92}$$

$$\leq (1 - \beta)n\Delta_{max} + 2KMN\zeta(2 - \varepsilon)\Delta_{max} \tag{93}$$

Finally, we can calculate the regret upper bound:

$$Reg(n) \leq n \cdot \alpha \cdot \beta \cdot \text{opt}_{\boldsymbol{\mu}} - \left( \alpha \cdot n \cdot \text{opt}_{\boldsymbol{\mu}} - Reg_u(n) - Reg_t(n) - Reg_s(n) \right) \tag{94}$$

$$= (\beta - 1) \cdot n \cdot \alpha \cdot \text{opt}_{\boldsymbol{\mu}} + Reg_u(n) + Reg_t(n) + Reg_s(n) \tag{95}$$

$$\leq \sum_{i \in [\![K]\!], \Delta^i_{min} \geq 0} \left( \ell_n(\Delta^{i,K_i}) \Delta^{i,K_i} + \int_{\Delta^{i,K_i}}^{\Delta^{i,1}} \ell_n(x) dx \right) + 2KMN\zeta(2 - \varepsilon)\Delta_{\max} + \tag{96}$$

$$KMN \max_{k \in [\![K_i]\!], i \in [\![K]\!]} \{ (\ell_n(\Delta^{i,k}))^\varepsilon \Delta^{i,k} \}. \tag{97}$$

## F   PROOF OF SWITCHING COST

We first give the following two lemmas.

**Lemma F.1**  *Let $\{a_n\}_{n \geq 0}$ be a sequence defined recursively by*

$$a_n = a_{n-1} + Na_{n-1}^\varepsilon, \quad \text{for } n \geq 1,$$

*with $a_0 > 0$, $N > 0$, and $0 < \varepsilon < 1$. Then there exists a constant $C > 0$ such that*

$$a_n \geq Cn^{\frac{1}{1-\varepsilon}}, \quad \forall n \geq n_0,$$

*for some integer $n_0 \geq 1$.*

Let $\gamma = \frac{1}{1-\varepsilon}$ and define $b_n = Cn^\gamma$. We will show by induction that $a_n \geq b_n$ for sufficiently large $n$.

Assume $a_n \geq b_n$ for some $n \geq n_0$, then

$$a_{n+1} = a_n + Na_n^\varepsilon \geq b_n + Nb_n^\varepsilon = Cn^\gamma + NC^\varepsilon n^{\varepsilon\gamma}.$$

Note that

$$b_{n+1} = C(n+1)^\gamma = Cn^\gamma \left(1 + \frac{1}{n}\right)^\gamma.$$

By the mean value theorem, for some $\xi \in (n, n+1)$,

$$(n+1)^\gamma - n^\gamma = \gamma \xi^{\gamma-1} \leq \gamma (n+1)^{\gamma-1}.$$

Thus,

$$b_{n+1} - b_n \leq C\gamma(n+1)^{\gamma-1}.$$

So, to ensure $a_{n+1} \geq b_{n+1}$, it suffices that

$$NC^\varepsilon n^{\varepsilon\gamma} \geq C\gamma(n+1)^{\gamma-1}.$$

Since $\varepsilon\gamma = \gamma - 1$, this reduces to:

$$NC^{\varepsilon-1} \geq \gamma \left(1 + \frac{1}{n}\right)^{\gamma-1}.$$

Observe that $\left(1 + \frac{1}{n}\right)^{\gamma-1} \leq 2^{\gamma-1}$ for all $n \geq 1$, so a sufficient condition is:

$$NC^{\varepsilon-1} \geq \gamma \cdot 2^{\gamma-1}.$$

This holds if we set

$$C = \left(\frac{\gamma \cdot 2^{\gamma-1}}{N}\right)^{\frac{1}{\varepsilon-1}} = \left(\frac{2^{\frac{\varepsilon}{1-\varepsilon}}}{N(1-\varepsilon)}\right)^{\frac{1}{\varepsilon-1}},$$

which is positive since $\varepsilon - 1 < 0$ and the base is positive. By choosing $n_0$ large enough so that $a_{n_0} \geq b_{n_0}$, the induction is complete.

**Lemma F.2** *By defining $t'$ as the start round of different phases, we have the following conclusion:*

$$C(n) \leq 2\mathbb{E}[\sum_{t'} \mathbb{I}(i_{t'} \neq i^*)]. \tag{98}$$

The above conclusion can be deduced as follows:

$$C(n) = \mathbb{E}[\sum_{t=1}^{n-1} \mathbb{I}(i_t \neq i_{t+1})] \tag{99}$$

$$= \mathbb{E}[\sum_{t' \neq 1} \mathbb{I}(i_{t'-1} \neq i_{t'})] \tag{100}$$

$$\leq \mathbb{E}[\sum_{t' \neq 1} 1 - \mathbb{I}(i_{t'-1} = i_{t'})] \tag{101}$$

$$= \mathbb{E}[\sum_{t' \neq 1} 1 - \sum_{j=1}^{K} \mathbb{I}(i_{t'-1} = i_{t'} = j)] \tag{102}$$

$$\leq \mathbb{E}[\sum_{t' \neq 1} 1 - \mathbb{I}(i_{t'-1} = i_{t'} = i_*)] \tag{103}$$

$$\leq \mathbb{E}[\sum_{t' \neq 1} \mathbb{I}(i_{t'-1} \neq i_*) + \mathbb{I}(i_{t'} \neq i_*)] \tag{104}$$

$$\leq 2\mathbb{E}[\sum_{t'} \mathbb{I}(i_{t'} \neq i_*)]. \tag{105}$$

During most of the time, UCB-based methods pull the best arm $i_*$. So we make some relaxation to the switching cost $C(n)$ to simplify our proof.

**Lemma F.3** *Initial concentrated exploration pull each (base) arm at least $\ln(n)$ times.*

If $T_i(t) \leq \ln(n)$, both $\sqrt{2\ln(n)/T_i(t)}$ and $\sqrt{3\ln(n)/2T_i(t)}$ are larger than 1, which contradicts with $\bar{\mu}'_{i,t} \leq 1$.

### F.1   PROOF OF THEOREM 5.4

For MSMR-UCB, when arm $i$ is selected for the $j$-th time, we define the length of the corresponding phase as $L_{i,j}$. Similar to the regret analysis, we decompose the total switching cost into three components. Specifically, let $t_{u,i}$, $t_{t,i}$, and $t_{s,i}$ denote the number of phases in which the sub-optimal arm $i$ is selected during the under-sampled stage, the transition stage, and the sufficiently sampled stage, respectively. The total switching cost is then bounded as:

$$C(n) \leq 2 \sum_{i \neq i_*} (t_{u,i} + t_{t,i} + t_{s,i}). \tag{106}$$

In the under-sampled stage, define $W_{i,s} = \sum_{j=1}^{s} L_{i,j}$ as the cumulative length of the first $s$ phases for arm $i$. According to the definition of near-optimal exploitation, we have:

$$W_{i,s} \geq W_{i,s-1} + N(W_{i,s-1})^{\varepsilon}. \tag{107}$$

Let $l_i$ denote the total number of pulls of arm $i$ during the under-sampled stage. By construction, $t_{u,i}$ satisfies:

$$W_{i,t_{u,i}+1} > l_i > W_{i,t_{u,i}}. \tag{108}$$

Applying Lemma F.3, if

$$\left(\frac{2^{\frac{\varepsilon}{1-\varepsilon}}}{N(1-\varepsilon)}\right)^{\frac{1}{\varepsilon-1}} < \ln(n), \tag{109}$$

we obtain:

$$L_{i,1} \geq \left(\frac{2^{\frac{\varepsilon}{1-\varepsilon}}}{N(1-\varepsilon)}\right)^{\frac{1}{\varepsilon-1}}. \tag{110}$$

Using Lemma F.1, we derive:

$$W_{i,t_{u,i}} \geq l_i \tag{111}$$

$$\Leftarrow \quad \left(\frac{2^{\frac{\varepsilon}{1-\varepsilon}}}{N(1-\varepsilon)}\right)^{\frac{1}{\varepsilon-1}} t_{u,i}^{\frac{1}{1-\varepsilon}} \geq \frac{8\ln(n)}{\Delta_i^2} \tag{112}$$

$$\Leftarrow \quad t_{u,i} \geq \left(\frac{8\ln(n)}{\Delta_i^2}\right)^{1-\varepsilon} \cdot \frac{2^{\frac{\varepsilon}{1-\varepsilon}}}{N(1-\varepsilon)}. \tag{113}$$

This contradicts $l_i < W_{i,t_{u,i}}$, so we conclude that:

$$t_{u,i} \leq \left(\frac{8\ln(n)}{\Delta_i^2}\right)^{1-\varepsilon} \cdot \frac{2^{\frac{\varepsilon}{1-\varepsilon}}}{N(1-\varepsilon)}. \tag{114}$$

In the transition stage, by definition, $t_{t,i} \leq 1$ for any arm $i$.

In the sufficiently sampled stage, the number of phases $t_{s,i}$ for each arm $i$ is upper bounded by $2MN \cdot \zeta(2-\varepsilon)$, which is a constant.

Combining all three stages, the total switching cost for MSMR-UCB satisfies:

$$C(n) \leq 2\sum_{i=1}^{K}(t_{u,i} + t_{t,i} + t_{s,i}) \tag{115}$$

$$\leq 2K + 4KMN \cdot \zeta(2-\varepsilon) + 2\sum_{i\neq i_*}\left(\frac{8\ln(n)}{\Delta_i^2}\right)^{1-\varepsilon} \cdot \frac{2^{\frac{\varepsilon}{1-\varepsilon}}}{N(1-\varepsilon)}. \tag{116}$$

## F.2 PROOF OF THEOREM 5.5

For MSMR-CUCB, when a super arm $S$ is selected and base arm $i = i_{S,t} \triangleq \arg\min_{j\in S} T_j(t)$ is chosen for the $j$-th time, we define the length of the corresponding phase as $L_{i,j}$. As in the regret analysis, we decompose the switching cost into three parts. Let $t_{u,i}$ and $t_{t,i}$ denote the number of phases in which a super arm containing base arm $i$ is selected during the under-sampled and transition stages, respectively. Let $t_s$ denote the number of phases in which any sub-optimal super arm is selected in the sufficiently sampled stage. The total switching cost is then bounded by:

$$C(n) \leq 2\left(t_s + \sum_{i=1}^{K}(t_{u,i} + t_{t,i})\right). \tag{117}$$

In the under-sampled stage, define $W_{i,s} = \sum_{j=1}^{s} L_{i,j}$ as the cumulative length of the first $s$ phases for base arm $i$. By the definition of near-optimal exploitation, we have:

$$W_{i,s} \geq W_{i,s-1} + N(W_{i,s-1})^{\varepsilon}. \tag{118}$$

Let $l_i$ denote the total number of pulls of base arm $i$ in the under-sampled stage. Then the total number of under-sampled phases $t_{u,i}$ satisfies:

$$W_{i,t_{u,i}+1} > l_i > W_{i,t_{u,i}}. \tag{119}$$

Applying Lemma F.3, if

$$\left( \frac{2^{\frac{\varepsilon}{1-\varepsilon}}}{N(1-\varepsilon)} \right)^{\frac{1}{\varepsilon-1}} < \ln(n), \tag{120}$$

then we obtain:

$$L_{i,1} \geq \left( \frac{2^{\frac{\varepsilon}{1-\varepsilon}}}{N(1-\varepsilon)} \right)^{\frac{1}{\varepsilon-1}}. \tag{121}$$

Using Lemma F.1, we derive:

$$W_{i,t_{u,i}} \geq \ell_n(\Delta^{i,K_i}) \tag{122}$$

$$\Leftarrow \quad \left( \frac{2^{\frac{\varepsilon}{1-\varepsilon}}}{N(1-\varepsilon)} \right)^{\frac{1}{\varepsilon-1}} t_{u,i}^{\frac{1}{1-\varepsilon}} \geq \frac{6\ln(n)}{(f^{-1}(\Delta^{i,K_i}))^2} \tag{123}$$

$$\Leftarrow \quad t_{u,i} \geq \left( \frac{6\ln(n)}{(f^{-1}(\Delta^{i,K_i}))^2} \right)^{1-\varepsilon} \cdot \frac{2^{\frac{\varepsilon}{1-\varepsilon}}}{N(1-\varepsilon)}. \tag{124}$$

This contradicts $l_i < W_{i,t_{u,i}}$, so we conclude that:

$$t_{u,i} \leq \left( \frac{6\ln(n)}{(f^{-1}(\Delta^{i,K_i}))^2} \right)^{1-\varepsilon} \cdot \frac{2^{\frac{\varepsilon}{1-\varepsilon}}}{N(1-\varepsilon)}. \tag{125}$$

In the transition stage, by definition, $t_{t,i} \leq K_i$ for each base arm $i$.

In the sufficiently sampled stage, the number of phases $t_s$ involving any sub-optimal super arm is at most $2KMN \cdot \zeta(2-\varepsilon)$, which is a small constant.

Therefore, the total switching cost of MSMR-CUCB is bounded by:

$$C(n) = 2\left( t_s + \sum_{i=1}^{K}(t_{u,i} + t_{t,i}) \right) \tag{126}$$

$$\leq 4KMN \cdot \zeta(2-\varepsilon) + 2\sum_{i=1}^{K} \left( \frac{6\ln(n)}{(f^{-1}(\Delta^{i,K_i})^2)} \right)^{1-\varepsilon} \cdot \frac{2^{\frac{\varepsilon}{1-\varepsilon}}}{N(1-\varepsilon)} + 2\sum_{i=1}^{K} K_i. \tag{127}$$

## G PROOF OF MARGINAL LOSS

In MAB setting, According to (3), the regret upper bound of UCB is

$$Reg^{opt}(n) = \sum_{i \neq i_*} \left( \frac{8\ln n}{\Delta_i} + (1 + \frac{\pi^2}{3})\Delta_i \right). \tag{128}$$

In CMAB setting, according to (7), the regret upper bound of CUCB is

$$\sum_{i \in [\![K]\!], \Delta_{min}^i \geq 0} \left( \ell_n(\Delta^{i,K_i})\Delta^{i,K_i} + \int_{\Delta^{i,K_i}}^{\Delta^{i,1}} \ell_n(x)\,dx \right) + (1 + \frac{\pi^2}{3}) \cdot m \cdot \Delta_{\max}. \tag{129}$$

In the greedy algorithm, it directly pulls each base(super) arm fixed times and then chooses the one with the largest empirical reward. In MAB setting, we have $K$ base arms, so $C^{opt}(n) = K$. In CMAB setting, the super arms may just contain 1 base arm, so the greedy algorithm at least to pull $K$ super arms to get the evaluation of all base arms, resulting in $C^{opt}(n) = K$. According to theorem 5.2 5.4 5.3 5.5 and the definition of marginal loss 4, we can get that:

$$R^{MSMR}(\lambda, n) = \mathcal{O}((\log n)^\varepsilon + \lambda(\log n)^{1-\varepsilon}) \tag{130}$$

For any other bandit algorithm X mentioned in table 1, it is easy to find that $Reg^X(n) - Reg^{opt}(n) = \mathcal{O}(Reg^X(n))$, so the marginal loss of algorithm X is $\mathcal{O}(Reg^X(n) + \lambda C^X(n))$. In most cases, $\lambda$ is constant, and setting $\varepsilon = 0.5$ yields the theoretical minimum marginal loss $\mathcal{O}(\lambda\sqrt{\log n})$

# H    PROOF ABOUT PREDICTIVE SELECTION

In MAB settings, for any arm $i$, once the agent chooses an arm $i_{t_1} \neq \hat{i}_{*,t_1}$ to explore at round $t_1$, the algorithm will repeatedly select $i_{t_1}$ in the subsequent phase regardless of the upper confidence estimation if the following inequality holds:

## H.1    PROOF OF LEMMA 5.6

In round $t_1$, we choose arm $i_{t_1}$ for exploration. After this phase, we define round $t_2 = t_1 + \gamma_{i_{t_1}}(t_1)$ and a subsequent round $t_3 = t_2 + \sum_{j \neq i_{t_1}, \hat{i}_{*,t_2}} \gamma_j(t_2)$.

If neither $\hat{i}_{*,t_2}$ nor $i_{t_1}$ is selected in any round $t \in [t_2, t_3]$, and Eq. 5 holds, we obtain the following sequence of inequalities:

$$\bar{\mu}_{\hat{i}_{*,t_2},t} = \sqrt{\frac{2\ln(t)}{T_{\hat{i}_{*,t_2}}(t)}} + \hat{\mu}_{\hat{i}_{*,t_2},t} \tag{131}$$

$$= \sqrt{\frac{2\ln(t)}{T_{\hat{i}_{*,t_2}}(t_2)}} + \hat{\mu}_{\hat{i}_{*,t_2},t_2} \tag{132}$$

$$\leq \sqrt{\frac{2\ln(t_3)}{T_{\hat{i}_{*,t_2}}(t_2)}} + \hat{\mu}_{\hat{i}_{*,t_2},t_2} \tag{133}$$

$$\leq \sqrt{\frac{2\ln(t_2)}{T_{i_{t_1}}(t_2)}} + \hat{\mu}_{i_{t_1},t_2} \tag{134}$$

$$\leq \bar{\mu}_{i_{t_1},t}, \tag{135}$$

which implies that the upper confidence bound of arm $i_{t_1}$ is always greater than or equal to that of $\hat{i}_{*,t_2}$ throughout the interval $[t_2, t_3]$. Consequently, if $\hat{i}_{*,t_2}$ is selected at any round $t \in [t_2, t_3]$, then arm $i_{t_1}$ must have already been selected at least once prior to $t$.

We now consider three exhaustive cases:

**Case 1: Arm $\hat{i}_{*,t_2}$ is selected in some round $t \in [t_2, t_3]$.**    From Eq. 135, we conclude that arm $i_{t_1}$ must have been selected at least twice before arm $\hat{i}_{*,t_2}$ is selected. Therefore, Lemma 5.6 holds in this case.

**Case 2.1: Arm $\hat{i}_{*,t_2}$ is not selected in $[t_2, t_3]$, but arm $i_{t_1}$ is.**    In this case, arm $i_{t_1}$ is selected at least twice before round $t_3$, while arm $\hat{i}_{*,t_2}$ is never selected. Thus, Lemma 5.6 also holds here.

**Case 2.2: Neither arm $\hat{i}_{*,t_2}$ nor arm $i_{t_1}$ is selected in $[t_2, t_3]$.** We further divide this into two subcases:

**Subcase 1:** If there exists an arm $i' \notin \{\hat{i}_{*,t_2}, i_{t_1}\}$ that is selected twice in the interval $[t_2, t_3]$, then this arm is selected at least twice before round $t_3$, and Lemma 5.6 is satisfied.

**Subcase 2:** If no such arm $i' \notin \{\hat{i}_{*,t_2}, i_{t_1}\}$ is selected more than once in $[t_2, t_3]$, then every arm in this set is selected at most once. However, since

$$t_3 = t_2 + \sum_{j \notin \{i_{t_1}, \hat{i}_{*,t_2}\}} \gamma_j(t_2),$$

it follows that each arm $j \notin \{i_{t_1}, \hat{i}_{*,t_2}\}$ is selected exactly once between $t_2$ and $t_3$. By Eq. 135, arm $i_{t_1}$ still maintains a higher UCB than $\hat{i}_{*,t_2}$ at round $t_3$, and hence the algorithm must select an arm $i' \neq \hat{i}_{*,t_2}$. Since every such arm has already been selected once from $t_1$ to $t_3$, this implies that one of them is selected twice. Thus, Lemma 5.6 is proved in this case as well.

## H.2 PROOF OF LEMMA 5.7

In round $t_1$, we choose a super arm $S_{t_1}$ for exploration. After this phase, we set $t_2 = t_1 + \gamma_{S_{t_1}}(t_1)$ and define a future round $t_3 = t_2 + \sum_{j \in [\![K]\!] \setminus (S_{t_1} \cup \hat{S}_{*,t_2})} \gamma_j(t_2)$.

We decompose the base arm set $[\![K]\!]$ into four disjoint subsets:

$$U_1 = S_{t_1} \cap \hat{S}_{*,t_2},$$
$$U_2 = S_{t_1} \setminus U_1,$$
$$U_3 = \hat{S}_{*,t_2} \setminus U_1,$$
$$U_4 = [\![K]\!] \setminus (S_{t_1} \cup \hat{S}_{*,t_2}).$$

By the monotonicity assumption, if Eq. (6) holds, then there exists $k_1 \in U_2$ and $k_2 \in U_3$ such that

$$\hat{\mu}_{k_1,t_2} + \sqrt{\frac{3\ln(t_2)}{2T_{k_1}(t_2)}} \geq \hat{\mu}_{k_2,t_2} + \sqrt{\frac{3\ln(t_3)}{2T_{k_1}(t_2)}}.$$

Otherwise, for any $k_1 \in U_2$, $k_2 \in U_3$, we have

$$\hat{\mu}_{k_1,t_2} + \sqrt{\frac{3\ln(t_2)}{2T_{k_1}(t_2)}} < \hat{\mu}_{k_2,t_2} + \sqrt{\frac{3\ln(t_3)}{2T_{k_1}(t_2)}},$$

and for any $k_3 \in U_1$,

$$\hat{\mu}_{k_3,t_2} + \sqrt{\frac{3\ln(t_2)}{2T_{k_3}(t_2)}} < \hat{\mu}_{k_3,t_2} + \sqrt{\frac{3\ln(t_3)}{2T_{k_3}(t_2)}}.$$

Now consider the ordered arrangements of base arms in $S_{t_1}$ and $\hat{S}_{*,t_1}$:

$$A = \{U_{1,1}, \ldots, U_{1,|U_1|}, U_{2,1}, \ldots, U_{2,|U_2|}\},$$
$$B = \{U_{1,1}, \ldots, U_{1,|U_1|}, U_{3,1}, \ldots, U_{3,|U_3|}\},$$

where $|\cdot|$ denotes the cardinality of the set and $U_{x,j}$ denotes the $j$-th smallest element of $U_x$. Then for any $j \in \{1, 2, \ldots, m\}$, we have $\bar{\mu}_{A_j,t_2} < \bar{\mu}_{B_j,t_3}$, which implies by the monotonicity assumption that

$$r(\hat{\boldsymbol{\mu}}_{t_2}, \boldsymbol{c}_{t_2}, S_{t_1}) < r(\hat{\boldsymbol{\mu}}_{t_2}, \boldsymbol{c}_{t_3}, \hat{S}_{*,t_2}),$$

contradicting Eq. (6).

If none of the base arms in $U_2 \cup \{k_2\}$ are selected during $t \in [t_2, t_3]$ and Eq. (6) holds, we derive the following:

$$\bar{\mu}_{k_2,t} = \hat{\mu}_{k_2,t} + \sqrt{\frac{3\ln(t)}{2T_{k_2}(t)}}$$

$$= \hat{\mu}_{k_2,t_2} + \sqrt{\frac{3\ln(t)}{2T_{k_2}(t_2)}}$$

$$\leq \hat{\mu}_{k_2,t_2} + \sqrt{\frac{3\ln(t_3)}{2T_{k_2}(t_2)}}$$

$$\leq \hat{\mu}_{k_1,t_2} + \sqrt{\frac{3\ln(t_2)}{2T_{k_1}(t_2)}}$$

$$\leq \bar{\mu}_{k_1,t}. \tag{136}$$

This indicates that for any super arm $S$ containing base arm $k_2$, there exists an alternative super arm $S' = S \setminus \{k_2\} \cup \{k_1\}$ such that

$$r_{\bar{\boldsymbol{\mu}}_t}(S) \leq r_{\bar{\boldsymbol{\mu}}_t}(S').$$

Since $k_2 \in \hat{S}_{*,t_2}$, this implies that there exists another super arm better than $\hat{S}_{*,t_2}$ in round $t \in [t_2, t_3]$ if no arm from $U_2 \cup \{k_2\}$ is selected.

**Case 1:** A super arm $S$ containing base arm $k_2$ is selected during $t \in [t_2, t_3]$.
By Eq. (136), it must be that a super arm containing $k_1$ has been selected before $S$ is selected. Since $k_1 \notin \hat{S}_{*,t_2}$, it is pulled at least twice before $t$, completing the proof of Lemma 5.7.

**Case 2.1:** No super arm containing $k_2$ is selected during $[t_2, t_3]$, but a super arm containing $i \in U_2$ is selected. Then $i$ is pulled twice before $t_3$, and $\hat{S}_{*,t_2}$ is never selected before $t_3$, completing the proof of Lemma 5.7.

**Case 2.2:** No super arm containing any element of $\{k_2\} \cup U_2$ is selected during $[t_2, t_3]$.
Let $S_1, \ldots, S_{K-|S_{t_1} \cup \hat{S}_{*,t_2}|}$ be the next super arms selected during $[t_2, t_3]$, and define the recursive update:

$$x_j = \gamma_{S_j}\left(t_2 + \sum_{k=1}^{j-1} x_k\right), \quad t_3' = t_2 + \sum_{j=1}^{K-|S_{t_1} \cup \hat{S}_{*,t_2}|} x_j.$$

Each $S_j \subseteq U_3 \cup U_4 \setminus \{k_2\}$, so each contains at least one element of $U_4$.

**Subcase 1:** If any base arm $i \in U_4$ is selected twice in $[t_2, t_3]$, the proof is complete.

**Subcase 2:** If no base arm $i \in U_4$ is selected more than once, then for each $j$,

$$x_j = \gamma_{S_j}(t_2 + \sum_{k=1}^{j-1} x_k) \leq \gamma_{\{k_2\} \cup S_j \setminus \hat{S}_{*,t_2}}(t_2).$$

Therefore, $t_3 \geq t_3'$. Since every $i \in U_4$ is selected once during $[t_2, t_3]$ and

$$\{k_1\} \cup \hat{S}_{*,t_2} \setminus \{k_2\}$$

has a higher estimated reward than $\hat{S}_{*,t_2}$ in round $t_3$, the algorithm selects another super arm. As all base arms not in $U_3 \setminus \{k_2\}$ are selected once starting from $t_1$, there exists at least one base arm $i' \notin \hat{S}_{*,t_2}$ that is pulled twice before $t_3$, completing the proof of Lemma 5.7.

# I  MORE EXPERIMENTS

Here we present the complete set of experiments.

## I.1  EXPERIMENT SETUP

Each experiment was conducted over 20 independent trials to ensure reliability, with $n = 100000$, $N = 1$, $M = 5$ for all bandit settings and $\alpha = 0.95$ , $\beta = 1$ for CMAB. The tests were performed on a macOS system equipped with an Apple M3 Pro processor and 18 GB of RAM.

**Baselines.** The algorithms used for comparison include UCB (3), phase-UCB(16), Batched Tsallis-INF(1), Batched Arm Elimination (13)in MAB settings and CUCB (6), phased-CUCB(16), B-FTRL(11) in CMAB setting.

**Regret, Switching Cost and Marginal Loss.** We set $\varepsilon = 0.5$ in MSMR and MSMR-P. We set $\gamma(t) = 0.01t$ and $\gamma(t) = 0.001t$ for Phase-UCB and Phase-CUCB. For B-FTRL, we set $a = 0.5$ and $b = 1$. For marginal loss, we set $R^A(\lambda, n) = \max(Reg^A(n) - Reg^{opt}(n), 0) + \lambda \max(C^A(n) - C^{opt}(n), 0)$and set $\lambda = 0.1$.

**Data Generation.** For simulation, in MAB settings, we set $K = 10$ and give a similar reward distribution which $\mu_i = 0.3 + 0.002 \times i$. In each round $t$, an arm $i_t$ is selected and the agent observes the outcome $X_{i_t, t}$. We conduct experiments on cascading bandits of the CMAB as an instance, where the objective is to select $m = 5$ items from a set of $K = 20$ to maximize the reward. We still give a very similar reward distribution which is $\mu_i = 0.3 + 0.002 \times i$. In each round $t$, a list $S_t = (a_{t,1}, \ldots, a_{t,m}) \subseteq [\![K]\!]$ is randomly selected. The outcome $X_{t,i}$ for each $i \in S_t$ is generated from a Bernoulli distribution with mean $\mu_i$. Given the ranked list $S_t$, if stopping at the $j_t$-th item, the observed outcomes are: $(X_{t,a_1}, \ldots, X_{t,a_k}) = (0, \ldots, 0, 1, x, \ldots, x)$, where the first $j_t - 1$ items are 0, the $j_t$-th item is 1, and the rest are unobserved ($x$). If the list is exhausted, the observed outcomes are: $(X_{t,a_1}, \ldots, X_{t,a_k}) = (0, 0, \ldots, 0)$. The reward is 1 for stopping and 0 for exhausting the list. The reward function can be written as $r(S_t; \boldsymbol{\mu}) = 1 - \prod_{i \in S_t}(1 - \mu_i)$.

For the real-world setting, we conduct experiments using the real-world Last.fm dataset (5), sourced from the Last.fm online music platform[1]. This dataset comprises 186,479 tag assignments, connecting 1,892 users to 17,632 artists. To model user preferences, following the existing works (10; 17), we derive feedback from ratings: if an item's rating exceeds 3, we assign a feedback value of 1; otherwise, it is 0. This binary feedback approach simplifies the representation of user preferences by distinguishing positive interactions from negative or neutral ones. Based on this feedback, we calculate the expected reward $\mu$ for each item in the Last.fm dataset. In MAB settings, we set $K = 10$ and randomly choose 10 similar distributions which $\mu_i \in [0.3, 0.32]$ is randomly sampled. In CMAB settings, We conduct experiments on cascading bandits as an instance, where the objective is to select $m = 5$ items from a set of $K = 20$ to maximize the reward. We still give 20 similar reward distributions which $\mu_i \in [0.3, 0.34]$ is randomly sampled.

## I.2  EXPERIMENTAL RESULTS

**Synthetic Data.** In Figures 2(a) and 3(a), we observe that the regret of MSMR and MSMR-P closely matches that of the standard baseline methods. In contrast, Batched Tsallis-INF and B-FTRL exhibit noticeably higher regret. Figures 2(b) and 3(b) further show that standard and phased methods suffer from a substantial number of switches, often exceeding several thousand. Batched Tsallis-INF and B-FTRL also incur a significant number of switches. In comparison, MSMR results in only 289 switches in the MAB setting and 432 in the CMAB setting, amounting to merely 2.1% of the switches incurred by UCB and 2.4% of those by CUCB. Moreover, MSMR-P achieves even greater savings, reducing switching to just 1.0% of UCB and 1.3% of CUCB—representing a nearly 50% reduction in switching cost compared to MSMR. These results highlight

---

[1] https://www.last.fm

the effectiveness of the *predictive selection* technique. In terms of marginal loss, figures 2(c) and 3(c) further shows that the MSMR framework achieves remarkably low loss compared to the best existing algorithm, significantly outperforming all other methods.

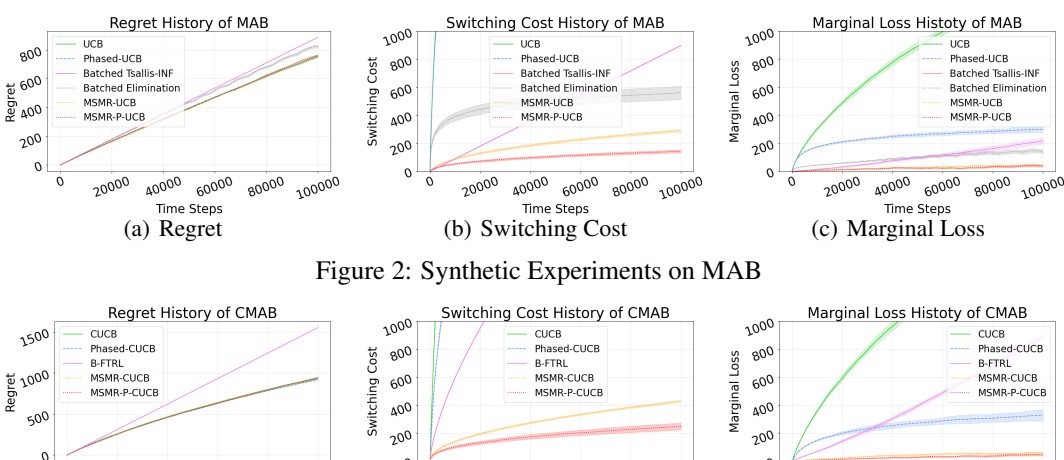

Figure 2: Synthetic Experiments on MAB

Figure 3: Synthetic Experiments on CMAB

**Real-world Data.** On real-world datasets, we find trends consistent with the simulation results. As shown in Figures 4(a) and 5(a), MSMR and MSMR-P achieve regret comparable to standard baselines, while Batched Tsallis-INF and B-FTRL incur noticeably higher regret. Figures 4(b) and 5(b) demonstrate that MSMR substantially reduces switching, with MSMR-P achieving the lowest cost overall. Finally, Figures 4(c) and 5(c) confirm that the MSMR framework maintains consistently low marginal loss, outperforming all existing methods.

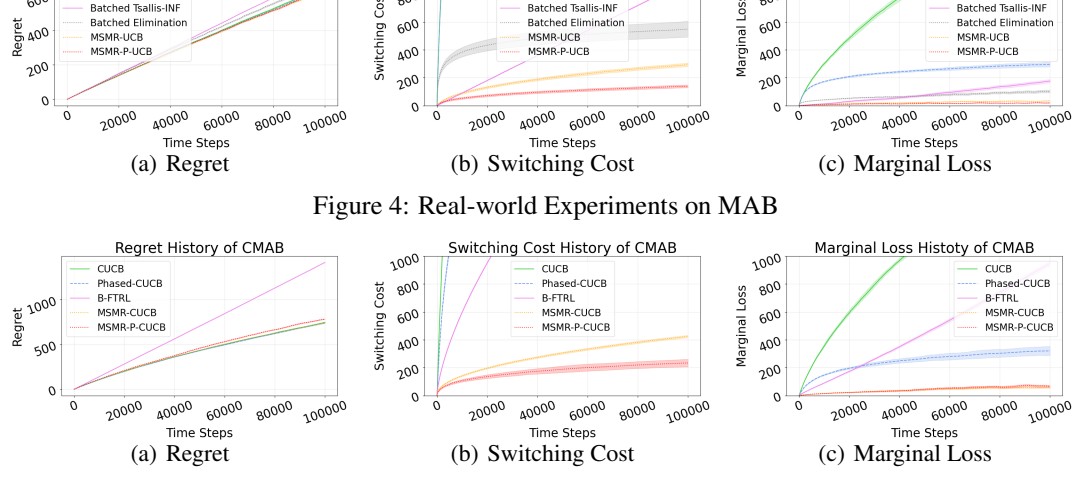

Figure 4: Real-world Experiments on MAB

Figure 5: Real-world Experiments on CMAB

## I.3 ABLATION STUDIES

In this section, we investigate the impact of various settings on the MSMR algorithm, including different arm gaps and different values of $\varepsilon$.

**Different Arm Settings.** For arm sets with larger reward gaps, identifying the optimal arm becomes relatively easier, leading to greater gains. Our primary interest, however, lies in scenarios with small gaps, where we aim to examine whether the algorithm can still secure competitive rewards while incurring only minimal switching cost. Accordingly, we design experiments with varying arm gaps for comparison. For MAB setting, we set $K = 10, \mu_i = 0.3 + 0.002(i - 1)$ in hard mode and $K = 5, \mu_i = 0.3 + 0.01(i - 1)$ in easy mode. For CMAB setting, we set $K = 20, m = 5, \mu_i = 0.3 + 0.002(i - 1)$ in hard mode and $K = 10, m = 3, \mu_i = 0.3 + 0.01(i - 1)$ in easy mode. The vertical axis *ratio* represents the percentage of regret and switching cost in the hard mode compared to the easy mode.

We observe that across different arm settings, the percentage change in regret from Easy to Hard mode remains consistent across algorithms. However, in terms of switching cost, Figure 6(a) shows that in the MAB setting, MSMR-UCB maintains stable performance, while MSMR-P-UCB experiences a much smaller increase than other algorithms, highlighting its advantage in handling arms with similar rewards. The performance of MSMR is further explained by the fact that, in this experiment, it switches only about 100 times, nearly reaching the theoretical lower bound. In the CMAB setting, as shown in Figure 6(b), regret across algorithms also remains relatively stable. Yet under the Hard mode, the increase in switching frequency for MSMR-based algorithms is minimal, with MSMR-P remaining almost unchanged.

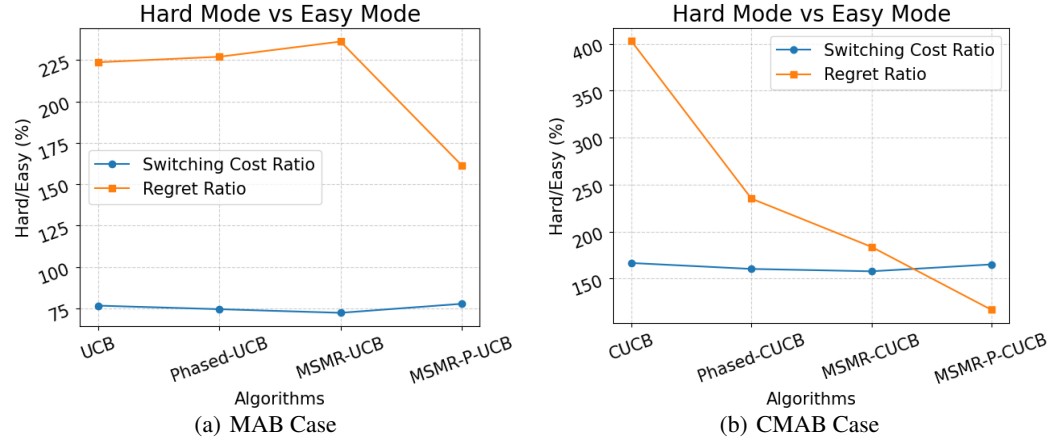

(a) MAB Case                    (b) CMAB Case

Figure 6: Different Arm Settings

**Different** $\varepsilon$. To further study the impact of the function $\gamma(t)$ on switching cost, we conduct a series of experiments by varying the parameter $\varepsilon$. We set $\varepsilon = 0.2 + 0.05 \times (j + 1)$ for $j = 1, 2, \ldots, 10$. As shown in Figures 7(a) and 7(b), both MSMR and MSMR-P experience reduced switching cost as $\varepsilon$ increases. When $\varepsilon \geq 0.3$, the switching cost remains consistently low for both algorithms. Notably, across all tested values of $\varepsilon$, MSMR-P consistently outperforms MSMR in terms of switching cost due to the design of the predictive selection technique, especially when $\varepsilon$ is small .

## J PHASED METHODS VS MSMR

In fact, the switching cost upper bound of Phased-UCB and Phased-CUCB is $\mathcal{O}(\log \log n)$. We take the MAB setting as an example to illustrate why the $\mathcal{O}(\sqrt{\log n})$ bound of our MSMR framework is preferable to

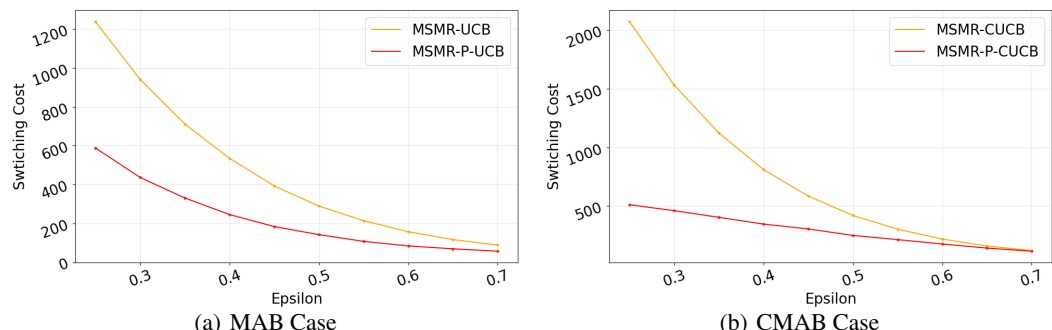

(a) MAB Case                    (b) CMAB Case

Figure 7: Different $\varepsilon$

$\mathcal{O}(\log \log n)$. The regret upper bound of Phased-UCB is $\mathcal{O}((1 + \kappa) \log n)$, and to control regret one typically selects a small $\kappa$. In our experiments, we set $\kappa = 0.01$.

Since Phased-UCB is a specific instance of the MSMR framework, we can similarly derive the upper bound on its switching cost. which is

$$C^{Phased}(n) = \frac{2}{\kappa} \sum_{i \neq i_*} (\ln(\frac{1}{\kappa}) + \max(2\ln(\kappa) + \ln\left(\frac{8\ln(n)}{\Delta_i^2}\right), 0) + 8K \tag{137}$$

According to 5.4,the switching cost upper bound of MSMR-UCB is:

$$C^{MSMR-UCB}(n) = 4KMN \cdot \zeta(2 - \varepsilon) + 2 \sum_{i \neq i_*} \left(\frac{8\ln(n)}{\Delta_i^2}\right)^{1-\varepsilon} \frac{2^{\frac{\varepsilon}{1-\varepsilon}}}{N(1 - \varepsilon)} + 2K. \tag{138}$$

Let $C^{Phased}(n) = C^{MSMR-UCB}(n)$, we can evaluate $n$ by the following expression:

$$n = \exp(\frac{\Delta_{\min}^2}{8} \exp(-\frac{B'}{C'} - \frac{1}{1 - \varepsilon} W_{-1}(-\frac{D'(1 - \varepsilon)}{C'} \exp(\frac{B'(1 - \varepsilon)}{C'})))), \tag{139}$$

where $W_{-1}(\cdot)$ is the lower branch of Lambert W Function,

$$B' = (K - 1)(\frac{\ln(\kappa)}{\kappa} - 2MN\zeta(2 - \varepsilon) + 3) - \frac{2}{\kappa} \sum_{i=1}^{K-1} \ln(\frac{\Delta_i}{\Delta_{\min}}), \tag{140}$$

$$C' = \frac{K - 1}{\kappa}, \quad D' = \frac{2^{\frac{\varepsilon}{1-\varepsilon}}}{N(1 - \varepsilon)} \sum_{i=1}^{K-1} (\frac{\Delta_{\min}}{\Delta_i})^{2-2\varepsilon}, \quad \Delta_{\min} = \min_{i \neq i_*} \Delta_i \tag{141}$$

It is worth noting that $n$ is a very large and complex expression. For example, in the main text experiment with $K = 10$, $\kappa = 0.01$, $\Delta_{\min} = 0.002$, $\varepsilon = 0.5$, $\mu_i = 0.3 + 0.002i$, $M = 5$, and $N = 1$, even such a small $\Delta_{\min}$ produces an extremely large $n$, with $n \geq 10^{500}$. Accordingly, MSMR maintains a theoretical advantage over Phased-UCB as long as $n \leq 10^{500}$ in our experiment, which already covers any conceivable practical application. Therefore, in realistic scenarios, MSMR can be regarded as superior to Phased-UCB, not only in terms of regret but also in terms of switching cost.

It is worth noting that $n$ is a very large and complex expression. For example, in the main text experiment with $K = 10$, $\kappa = 0.01$, $\Delta_{\min} = 0.002$, $\varepsilon = 0.5$, $\mu_i = 0.3 + 0.002i$, $M = 5$, and $N = 1$, even such a small

$\Delta_{\min}$ yields an extremely large $n$, with $n \geq 10^{500}$. Hence, MSMR maintains a theoretical advantage over Phased-UCB whenever $n \leq 10^{500}$ in our experiment, which covers all practical applications. Therefore, in realistic settings, MSMR can be regarded as superior to Phased-UCB, both in terms of regret and switching cost.